Journal of
open psychology data

# Understanding Psychological Responses to the COVID-19 Pandemic Through ESM Data: The EMOTIONS Project

DATA PAPER

ELINA RYVKINA

LARA KROENCKE

KATHARINA GEUKES

JULIAN SCHARBERT

MITJA D. BACK

*Author affiliations can be found in the back matter of this article

]u[ ubiquity press

## ABSTRACT

To examine psychological responses to the COVID-19 pandemic, we conducted two large experience sampling (ESM) studies (Study 1: $N_{\text{ESM participants}}$ = 327 and $N_{\text{ESM reports}}$ = 29,512, Study 2: $N_{\text{ESM participants}}$ = 2,272 and $N_{\text{ESM reports}}$ = 64,810). Each study subsumed two 14-day ESM waves that took place before and during (Study 1) or during and after (Study 2) the first nationwide COVID-19 lockdown in Germany. We describe the assessed variables on the trait (e.g., personality) and state (e.g., momentary emotions) level. All data sets and codebooks are shared on osf.io/6kzx3/. The EMOTIONS data are open to collaboration and inclusion in reviews/meta-analyses.

**CORRESPONDING AUTHORS:**

**Elina Ryvkina**

University of Münster, Department of Psychology, Fliednerstr. 21, 48149 Münster, Germany

elina.ryvkina@uni-muenster.de

**Mitja Back**

University of Münster, Department of Psychology, Fliednerstr. 21, 48149 Münster, Germany

mitja.back@uni-muenster.de

**KEYWORDS:**
COVID-19; coronavirus; pandemic reactions; emotional states; experience sampling

**TO CITE THIS ARTICLE:**
Ryvkina, E., Kroencke, L., Geukes, K., Scharbert, J., & Back, M. D. (2023). Understanding Psychological Responses to the COVID-19 Pandemic Through ESM Data: The EMOTIONS Project. *Journal of Open Psychology Data,* 11: 6, pp. 1–30. DOI: https://doi.org/10.5334/jopd.83

# 1. BACKGROUND

On March 11, 2020, Tedros Adhanom Ghebreyesus, the current director-general of the World Health Organization (WHO), declared COVID-19 a global pandemic (WHO, 2020). In his speech, he pointed out that the measures taken to curb the spread of the virus were deeply perturbing the health, social, and economic sectors of every affected country. In addition, researchers and laypeople alike have emphasized the individual and psychological impacts of the pandemic.[1] Initial research showed that COVID-19 has shaped and continues to shape people's cognitions, emotions, and (social) behaviors (e.g., Arora et al., 2022; Rudert et al., 2021).

Importantly, the crisis' impact is not universal, as individuals' reactions to it vary. Thus, the roles of interindividual differences in people's cognitive, emotional, and behavioral responses to the pandemic have been and remain subject to intensive exploration (e.g., Bacon et al., 2022; Zettler et al., 2022). For instance, agreeableness and conscientiousness predicted more positive attitudes toward spread-mitigating policies (e.g., Rammstedt et al., 2022). On the contrary, antagonistic personality traits (e.g., Dark Tetrad traits) were associated with greater disapproving views of government (Ścigała et al., 2021), impeded compliance with health recommendations (e.g., Rammstedt et al., 2022; Ścigała et al., 2021; Zettler et al., 2022), and diminished psychological well-being (Bacon et al., 2022; Kroencke et al., 2020).

However, existing research predominantly employed retrospective[2] assessments to measure psychological responses to the pandemic (Sterl et al., 2022; for two illustrative examples, see Lazarević et al., 2021; Peitz et al., 2021). Retrospective measures have several benefits pertaining to their practicality, such as cost and time efficacy (Paulhus & Vazire, 2007). Yet, they may be subject to memory biases (Larson & Csikszentmihalyi, 1983; Scollon et al., 2009; Raphael, 1987) and tend to obscure the intraindividual, everyday dynamics of people's cognitive, emotional, and behavioral states (e.g., Sterl et al., 2022).

To overcome these limitations and uncover how people think, feel, and behave in their daily lives, researchers can employ *experience sampling methodology* (ESM; for one of its earliest applications, see Brandstaetter, 1983). In ESM studies, participants receive multiple short surveys per day assessing their momentary experiences and behaviors (e.g., Larson & Csikszentmihalyi, 1983; Scollon et al., 2009). Thereby, ESM generates fine-grained and ecologically valid data on the frequency, intensity, and variability of psychological (i.e., cognitive, emotional, and behavioral) states and situational variables (e.g., the spatial and temporal context; Csikszentmihalyi & Larson, 1987; Funder, 2015; Scollon et al., 2009). In addition, studies employing ESM typically include surveys administering trait measures before and/or after the ESM phase (Beal, 2015). This allows researchers to analyze how (relatively) stable interindividual differences on the trait level are linked to (a) the average level of psychological states and (b) intraindividual state dynamics (that is, to how much and how fast states vary across time and how strongly they are related to each other within persons; e.g., Conner et al., 2009; Scollon et al., 2009).

With the EMOTIONS project, we draw on the aforementioned unique strengths of ESM studies to substantiate an ever-growing field of psychological COVID-19 research. We aim to gain a better understanding of psychological responses to the pandemic and (inter- as well as intra-) individual differences therein. To this end, we conducted two multi-wave ESM studies before, during, and after the first nationwide COVID-19 lockdown in Germany (Federal Ministry of Health, 2022). Therefore, we collected trait data on socio-demographic variables, personality traits, well-being, political attitudes, as well as trait-level COVID-19-related cognitions, emotions, and behaviors (henceforth: *trait measures*). Additionally, using ESM, we assessed momentary (partly COVID-19-related) cognitive, emotional, and behavioral states pertaining to social interactions or non-social activities (henceforth: *state measures*) alongside their situational context.

Insights from the EMOTIONS data can help respond to issues of great societal and political relevance. What made certain individuals (seemingly) resilient to impediments in their psychological well-being during the pandemic? Why did some people engage in social distancing, whilst others refrained from doing so? Who stockpiled daily goods, and who did not? Answers to these questions could practically inform politicians, health officials, and the public, guiding them and us through the daunting endeavor of coping with this and future global crises.

# 2. METHODS

## 2.1 STUDY DESIGN

The EMOTIONS project encompassed two ESM studies, one mostly with university students (henceforth: *Study 1*) and the other with members of the general population (henceforth: *Study 2*). Each study included two waves to cover different stages of the pandemic (for details, see Section 2.2). All waves shared a common procedure with three phases of data collection and were programmed in formr version 0.18.3, an open-source software for complex online surveys (Arslan et al., 2020). Start and end dates could vary between participants within each study wave's duration. However, every participant who properly finished a study wave completed 16 assessment days in total. Their purpose and content are broken down in the following.

In the first phase of data collection (i.e., day one), participants completed an initial, cross-sectional trait survey. Upon agreeing to the conditions of participation and generating a participant code, they provided information on multiple interindividual-difference variables, including socio-demographic variables (e.g., gender, occupational status, household size), personality traits (e.g., Big Five, narcissism), well-being (e.g., loneliness), political attitudes (e.g., political orientation, conspiracy mentality), and/or global experiences during the COVID-19 pandemic (e.g., risk perceptions, policy evaluations, behavioral changes; for a detailed overview of which measures were administered when, see Section 2.5 of this paper and/or the codebooks on the Open Science Framework [OSF]: osf.io/6kzx3/).

On the following day (i.e., day two), participants entered the second phase of data collection—a longitudinal, 14-day ESM phase, in which up to six short state surveys were completed per day. An email including a personalized access link prompted participants to complete each ESM survey.[3] These emails were sent at random times throughout the day, but within the bounds of every participant's individually preferred time window. Preferences for the earliest daily start time and the latest daily end time were specified at the end of the initial trait survey separately for weekdays and weekends. A possible start time could be between 4 am and 12 pm, and a possible end time was ought to be at least 10 hours after the specified start time.

Time lags between two consecutive surveys were programmed to be at least 40 min long. Participants had 45 min to complete each survey after receiving the email prompt. If they did not respond within 20 min, a reminder email was issued. Each ESM survey assessed the situational context of the participants' most recent, at least 5-min-long social interaction (e.g., number and type of interaction partners) or non-social activity (e.g., type of activity) as well as their momentary cognitive, emotional, and behavioral states.

The third and concluding phase of data collection followed a day after the final ESM day (i.e., day 16). There, participants responded to a final trait survey that was similar to the initial survey regarding its design and administered measures. A project timeline (incl. focal COVID-19 events alongside abbreviations to establish an unequivocal terminology) is presented in Figure 1. Table 1 presents a high-level overview of all measures that were administered per phase of data collection.

## 2.2 TIME OF DATA COLLECTION

In this section, we report on each study wave's data collection period besides relevant, COVID-19-related events in which it was embedded. The first wave of Study 1 (henceforth: Study 1 Wave 1 or simply S1W1[4]) was released on January 13, 2020, and ended on April 18, 2020. Thus, data collection started before the onset of the coronavirus crisis in Germany, which was marked by the first confirmed case in Bavaria on January 27, 2020 (Federal Ministry of Health, 2022). S1W1 focused on the

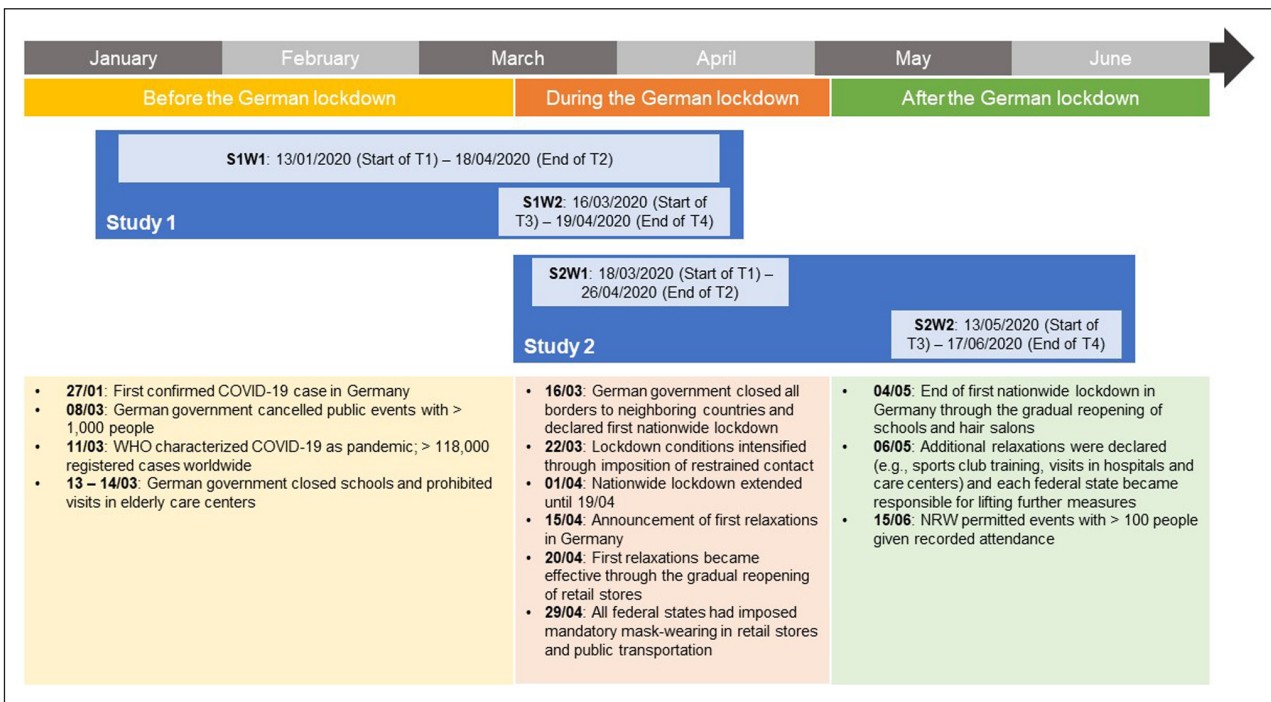

**Figure 1** EMOTIONS Project Timeline Including Terminology and Central COVID-19 Events.

*Note*: This table provides an overview of the entire EMOTIONS project, including both studies (Study 1 and 2) and each study's waves. S1W1 = Study 1 Wave 1 (all other waves are abbreviated analogously), T1 = time point 1 (initial trait survey of both studies' respective first wave), T2 = time point 2 (final trait survey of both studies' respective first wave), T3 = time point 3 (initial trait survey of both studies' respective second wave), T4 = time point 4 (final trait survey of both studies' respective second wave), NRW = North Rhine-Westphalia. Date format: DD/MM/YYYY. For study-wave-specific timelines, see each study wave's respective codebook on osf.io/6kzx3/.

| PHASE OF DATA COLLECTION | CONSTRUCT | MEASURE |
|---|---|---|
| Initial and final trait surveys | Basic personality traits | • Study 1: Interpersonal Adjective Scales (IAL)<br>• Study 2:<br> ○ Big Five Inventory-2-S (BFI-2-S)<br> ○ Honesty-Humility (subscale from the HEXACO-60) |
| | Narcissism | • Study 1:<br> ○ Narcissistic Admiration and Rivalry Questionnaire (NARQ)<br> ○ Hypersensitive Narcissism Scale (HSNS)<br>• Study 2: Narcissistic Admiration and Rivalry Questionnaire Short Scale (NARQ-S) |
| | Self-esteem | • Study 1: Rosenberg Self-Esteem Scale (RSES)<br>• Study 2: Single self-esteem item from the RSES |
| | Loneliness | • Study 1: *not assessed*<br>• Study 2: UCLA Loneliness Scale (ULS) |
| | Political orientation | • Study 1: *not assessed*<br>• Study 2: Single political orientation item |
| | Conspiracy mentality | • Study 1: *not assessed*<br>• Study 2: Conspiracy Mentality Questionnaire (CMQ) |
| | C19-related risk estimations | Self-generated measures |
| | C19-related worries | |
| | C19-related behavioral evaluations | |
| | C19-related behavioral changes | |
| | C19-related stockpiling | |
| | C19-related policy evaluations | |
| | Exposure to the coronavirus | |
| | C19-related personal restrictions | |
| ESM (state) surveys | Interaction-specific measures | • Type of activity during interaction<br>• Mode of communication<br>• Number of interaction partners<br>• Relationship to each interaction partner<br>• Behavioral states during the interaction (e.g., dominance, warmth) [a]<br>• Perceptual states during the interaction (e.g., status perceptions) [a]<br>• Emotional states immediately after the interaction (e.g., anger, sadness) [a] |
| | Non-social-activity-specific measures | • Type of activity<br>• Mode of activity<br>• C19-specific activities [a]<br>• Perceptual states during the activity (e.g., pleasure, boredom) [a]<br>• Emotional states during the activity (e.g., success, insecurity) [a] |
| | Interaction-unspecific measures | Overall affect valence and arousal |
| | Interaction-unspecific, C19-related worries [a] | • Self-related worries<br>• Society-related worries<br>• Other-related worries (pertaining to family members, partner, close friends, and wider social environment) |

**Table 1** High-Level Overview of Measures per Phase of Data Collection.

*Note*: This table lists all measures administered in the initial and final trait surveys (i.e., T1 and T2 in S1W1 and S2W1; T3 and T4 in S1W2 and S2W2) as well as the ESM surveys per study wave (e.g., S1W1 [Study 1 Wave 1; all other study waves are abbreviated analogously]). C19 = COVID-19. Each measure is assigned to the construct it is supposed to assess. The initial and final trait surveys incorporated trait measures only, whereas the ESM surveys exclusively administered state measures. For a comprehensive delineation of every measure included in the EMOTIONS project (alongside its respective German and English source), please refer to Section 2.5 and/or the study-wave-specific codebooks (osf.io/6kzx3/).

[a] Some COVID-19-related behavioral, perceptual, and emotional states were added in Study 2.

relationships between three aspects of narcissism (Back, 2018) and momentary status perceptions as well as behavioral and emotional states (Kroencke, Kuper, et al., 2022). To examine how these states were affected by the COVID-19 pandemic and to gain a better understanding of people's trait-level pandemic-related thoughts,

emotions, and behaviors, we launched additional study waves.

S1W2 was released on March 16, 2020, and ended on April 19, 2020. On March 11, 2020, just before the start of S1W2, the WHO had characterized COVID-19 as a pandemic, with more than 118,000 registered infection cases worldwide (WHO, 2020). Simultaneously, national events were evolving fast. To protect risk groups (i.e., the elderly and chronically ill), the German government had cancelled public events with more than 1,000 people (March 8, 2020), closed (nursery) schools, and prohibited the visit of elderly care centers (March 13 – 14, 2020) in most federal states (Tagesschau, 2020a). On March 16, 2020, the launch day of S1W2, Germany closed all its borders to neighboring countries (with some exceptions pertaining to commute, commodity transportation, and returning citizens) and declared the first nationwide lockdown (WirtschaftsWoche, 2022). The latter implied that, across federal states, non-essential retail stores as well as indoor public places (e.g., bars, clubs) were closed and public gatherings (in leisure facilities, educational institutions, places of worship, etc.) prohibited. Additional restrictions could be enforced by the federal states, including regulated visits to hospitals or the introduction of hygiene measures in restaurants and hotels (The chancellor and the heads of government of the states, 2020a). Lockdown conditions were further intensified on March 22, 2020. Although no rigorous confinement was imposed (in contrast to other European countries such as France or Spain), contact was extensively restricted. Specifically, meeting more than a single person from another household was forbidden, keeping at least 1.5 meters distance to other people was strongly encouraged, and restaurants as well as other facilities requiring close body contact (e.g., hairdressers, cosmetic studios) were shut down. However, people were still allowed to go to work, provide (emergency) care, shop for groceries, visit doctors, attend necessary appointments, sit exams, and practice individual sports outside (The chancellor and the heads of government of the states, 2020a). On April 1, 2020, the lockdown was extended until April 19, 2020. First relaxations pertaining to the gradual reopening of retail stores (April 20), hair salons (May 4), and schools (May 4) were announced on April 15, 2020. Nonetheless, large events were banned until August 31, 2020; and from April 29, 2020, onwards, all federal states had imposed mandatory mask-wearing in retail stores and public transportation (Tagesschau, 2020b).

S2W1 was released on March 18, 2020, and ended on April 26, 2020. Thus, it shared most of its COVID-19-related context with S1W2. S2W2 was a stand-alone study wave, launching on May 13, 2020, and ending on June 17, 2020. Prior to its release, the end of the first German nationwide lockdown was marked by the gradual reopening of schools and hair salons on May 4, 2020 (WirtschaftsWoche, 2022). Two days later,

additional relaxations were declared by the federal and state governments, including continued school and store reopening, outdoor sports club training, or the permission for a single person to visit patients in hospitals and elderly in care centers. Moreover, federal states were to decide individually about lifting further measures (e.g., reopening of universities, restaurants, hotels, or gyms) based on the development of infection rates. For instance, during data collection for S2W2, the state of North Rhine-Westphalia lifted the 14-day quarantine rule for people returning from neighboring countries (May 15, 2020) or permitted events with more than 100 people given their organizers' commitment to tract attendance (June 15, 2020; State chancellery of North Rhine-Westphalia, 2020). Notwithstanding such favorable development, central restrictions (e.g., meetings limited to multiple members of two households only, mask-wearing) remained intact, and local reactions to increasing infection rates had to be contingent and rapid (The chancellor and the heads of government of the states, 2020b).

## 2.3 LOCATION OF DATA COLLECTION

All study waves were conducted online. Based on our recruitment strategies, we mainly collected data for Study 1 among students at the University of Münster, Germany, and data for Study 2 was mostly provided by participants living in the Münsterland region. For further details on recruitment, see Section 2.4.1.1.

## 2.4 SAMPLING, SAMPLE, AND DATA COLLECTION

### 2.4.1 Sampling Strategy

#### 2.4.1.1 Recruitment

For S1W1, participants were recruited via advertisement in lectures as well as posters and flyers distributed at the University of Münster (e.g., Institute of Psychology, library, cafeteria) and in other public places (e.g., city center). Moreover, announcements were made on social media (e.g., Facebook groups) and internet forums. Having provided their email address to participate in the ESM phase of data collection, participants from S1W1 were recontacted via email to engage them for the second wave of Study 1. To recruit a German convenience sample for S2W1, we dispatched (social) media announcements (e.g., press release of the University of Münster, newsletter of the general student committee). For S2W2, participants from S2W1 were reinvited via email. We recruited additional participants via Facebook ads.

#### 2.4.1.2 Compensation

There were four types of compensation strategies pursued in the EMOTIONS project. Across all EMOTIONS waves, participants received (1) comprehensive, personalized feedback on their emotional well-being in everyday social life that was sent out after the respective

data collection period ended. This feedback summarized every participant's (a) positive and negative affective states in different contexts (e.g., social interactions vs non-social activities), (b) trajectories of positive and negative affective states across the 14-day ESM phase (comparing trajectories before and during the COVID-19 pandemic in feedback for Study 1, if participants completed both waves of Study 1), and (c) worries regarding the outbreak of the pandemic. Some individual feedback was compared to sample means. Further details on feedback timing and content are presented in the Supplemental Material.

In addition, in Study 1, students from the University of Münster were eligible to obtain (2) course credits. Specifically, in S1W1, six course credits could be acquired for completing at least 50% of the state surveys and four extra course credits for completing more than 80% of the state surveys (i.e., as a bonus for regular completion), leaving student participants with up to 10 course credits in total. In S1W2, completing more than 80% of the state surveys yielded five course credits.

Finally, participants in S1W1 could (3) enter a lottery for gift vouchers (25 Amazon vouchers worth EUR 50 each), whereas participants in S1W2 were (4) financially compensated, receiving EUR 10 if they completed state surveys for one week and EUR 30 for a completion rate of more than 80%.

### 2.4.2 Data Exclusion

We excluded data from the *trait data set* (subsuming data from trait surveys only) and the *state data set* (combining data from trait and state surveys)[5] of every EMOTIONS wave based on the following criteria:

**(a)** Disagreement with the conditions of participation (i.e., statements on anonymity, use of data, voluntary nature of participation)
**(b)** Missing data on all relevant variables (i.e., participants provided only a participant code but no further information)
**(c)** Participants under the minimum required age of 16 years
**(d)** Test users (i.e., data generated by members of our research group)
**(e)** Duplicates (i.e., participants who created multiple accounts)

Regarding criterion (a), only a single response option was provided, namely "I agree to the conditions of participation". Thus, those who did not tick this box automatically refused to grant their informed consent and could not proceed with the survey. Regarding criterion (e), we identified duplicates based on both email addresses and participant codes. Whenever participants created multiple accounts simultaneously (i.e., in parallel study waves or within the same study wave) and used

more than one account in the ESM phase, all accounts were deleted. If they used only a single account in the ESM phase, this account was retained, while the rest was excluded. Whenever participants created multiple accounts consecutively (i.e., creating a second account after completing the ESM phase with the first account), we retained the first or only account that was employed in the ESM phase.

Finally, we applied up to three additional data exclusion criteria uniquely to each state data set: (f) expired (i.e., incomplete) state surveys, (g) state surveys completed too close to each other (i.e., less than 40 min after the previous survey), and (h) technical error specific to S1W1, which lead to state surveys being sent out during the night on January 14, 2020.

Exclusion criteria were applied in the aforementioned order. Table 2 shows the number of participants excluded per data exclusion criterion in the trait data set of every EMOTIONS wave as well as the number of participants and state reports (i.e., completed state surveys) excluded per data exclusion criterion in the state data set of every EMOTIONS wave.

### 2.4.3 Descriptive Statistics on ESM Survey Completion

Over the course of a study wave's total duration, the number of ESM reports (i.e., completed ESM surveys) varied within participants. In Study 1, most ESM data was collected between January 14, 2020, and March 1, 2020[6] (S1W1), as well as between March 17, 2020, and April 2, 2020 (S1W2). In Study 2, we acquired most ESM data between March 19, 2020, and April 14, 2020 (S2W1), as well as between May 14, 2020, and June 4, 2020 (S2W2). Figures 2 to 5 portray the longitudinal trajectories of the daily number of ESM reports and corresponding daily number of participants per study wave. Moreover, Table 3 shows study-wave-specific means, medians, standard deviations, and ranges for the number of ESM reports as well as completed ESM days per participant.

### 2.4.4 Data Merging

As a result of reinviting participants from each study's first wave to participate in the second wave, Wave 1 and 2 (of Study 1 and 2, respectively) yielded relatively similar samples (for study-wave-specific details, see Table 5 in Section 2.4.5). To exploit this feature and create most comprehensive data sets, we merged the trait data sets (and state data sets, respectively) from both waves per EMOTIONS study. Participant accounts were merged based on identical emails and/or identical participant codes.[7] To increase the number of participants in the merged data sets, we included participants who completed either the first or second wave of the respective study only (e.g., either S1W1 or S1W2 from Study 1).[8]

Table 4 summarizes the number of observations (i.e., participants and—if applicable—state reports)

| STUDY | | | STUDY 1 | | | | | | STUDY 2 | | | | | | |
|---|---|---|---|---|---|---|---|---|---|---|---|---|---|---|---|
| STUDY WAVE | | | S1W1 | | | S1W2 | | | S2W1 | | | S2W2 | | | |
| DATA SET | | | TRAIT | STATE | PARTICIPANTS STATE REPORTS | TRAIT | STATE | PARTICIPANTS STATE REPORTS | TRAIT | STATE | PARTICIPANTS STATE REPORTS | TRAIT | STATE | PARTICIPANTS STATE REPORTS | |
| | | | PARTICIPANTS | PARTICIPANTS | | PARTICIPANTS | PARTICIPANTS | | PARTICIPANTS | PARTICIPANTS | | PARTICIPANTS | PARTICIPANTS | | |
| Number of observations before data exclusion | | | 2,938 | 2,938 | 20,639 | 1,480 | 1,480 | 13,590 | 12,016 | 12,016 | 53,792 | 6,004 | 6,004 | 29,969 | |
| Data exclusion criterion | | | | | | | | | | | | | | | |
| (a) Disagreement with the conditions of participation [a] | | | 2,482 | 2,482 | 2,482 | 1,242 | 1,242 | 1,242 | 8,941 | 8,941 | 8,941 | 4,314 | 4,314 | 4,314 | |
| (b) Missing data on relevant variables | | | 52 | 52 | 52 | 23 | 23 | 23 | 470 | 470 | 470 | 270 | 270 | 270 | |
| (c) Under minimum age | | | 0 | 0 | 0 | 0 | 0 | 0 | 0 | 0 | 0 | 0 | 0 | 0 | |
| (d) Test users | | | 2 | 2 | 37 | 1 [b] | 1 [b] | 47 [b] | 1 | 1 | 43 | 0 | 0 | 0 | |
| (e) Duplicates | | | 49 | 49 | 223 | 11 | 11 | 94 | 89 | 89 | 269 | 66 | 66 | 66 | |
| (f) State surveys erroneously dispatched during the night* | | | | 38 | 65 | | | | | | | | | | |
| (g) Expired state surveys* | | | | 2 | 269 | | 10 | 136 | | 870 | 3,072 | | 440 | 1,499 | |
| (h) State surveys dispatched too close to each other* | | | | 0 | 47 | | 0 | 0 | | 0 | 3 | | 0 | 4 | |
| Total number of excluded observations | | | 2,585 | 2,625 | 3,175 | 1,277 | 1,287 | 1,542 | 9,501 | 10,371 | 12,798 | 4,650 | 5,090 | 6,153 | |
| Total number of retained observations after data exclusion | | | 353 | 313 | 17,464 | 203 | 193 | 12,048 | 2,515 | 1,645 | 40,994 | 1,354 | 914 | 23,816 | |

**Table 2** Number of Excluded Observations in the Trait and State Data Sets per EMOTIONS Study Wave.

*Note:* This table presents the criterion-specific and total number of observations removed from the trait and state data sets of the EMOTIONS project as well as each data set's total number of retained observations after data exclusion. S1W1 = Study 1 Wave 1 (all other waves are abbreviated analogously). Exclusion criteria were applied in the order in which they are presented in this table. This means that for instance, duplicates (i.e., criterion [e]) were excluded from a data set to which all previous data exclusion criteria (i.e., [a] to [d]) had already been applied. Exclusion criteria that were uniquely applied to state data are marked with an asterisk (*). Age (just as any other socio-demographic variable) was not assessed in S1W2. Hence, criterion (c) could not be applied to the trait and state data set of S1W2. Criterion (f) was relevant for S1W1 only. Note that participants who were excluded according to criteria (a), (b), and (c) could not participate in the ESM phase of data collection. Consequently, from each state data set, there were as many (empty) state reports as participants deleted on these three criteria (e.g., 2,842 participants and state reports—i.e., empty rows in the state data set—were excluded on criterion [a] in S1W1).

[a] Participants could not deliberately disagree with the conditions of participation. Instead, not checking the box "I agree to the conditions of participation" (i.e., the only response option given on the consent item) lead to being excluded from continued study completion.

[b] There were no test users in S1W2. Instead, one participant explicitly requested to be excluded from the data sets.

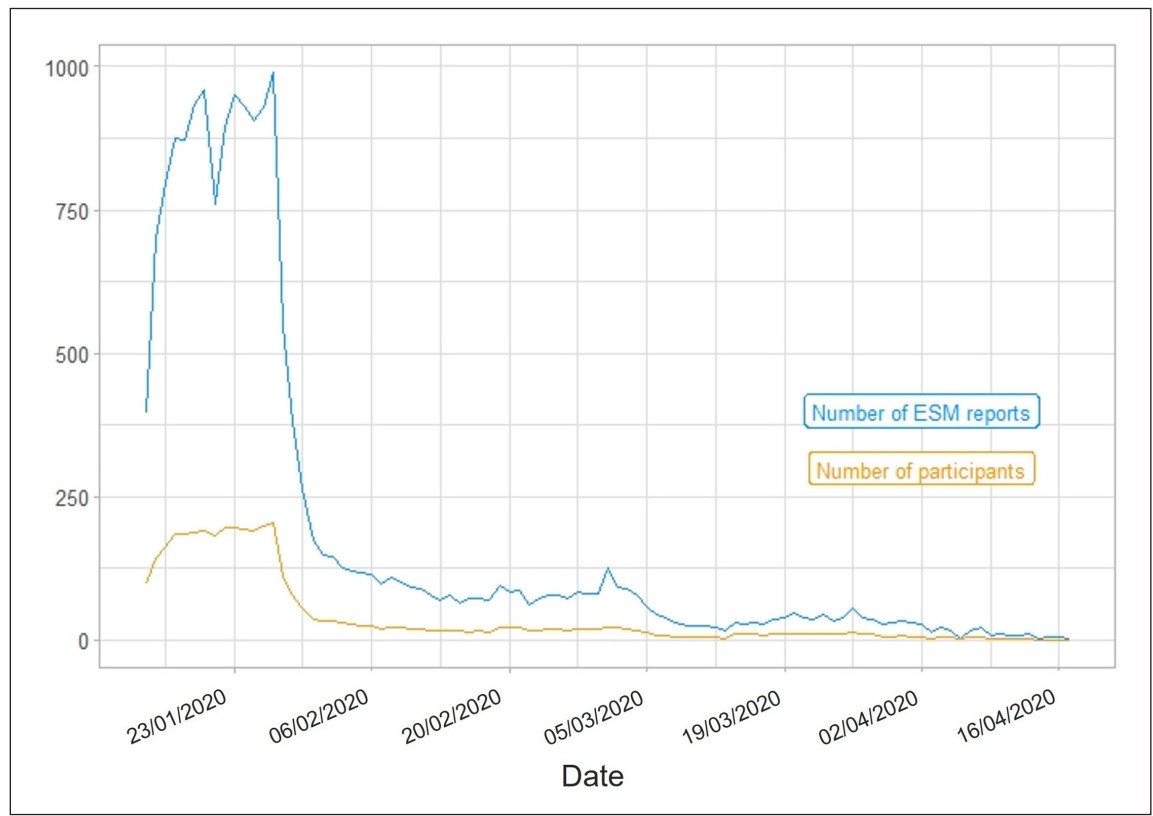

**Figure 2** Longitudinal Trajectories of Daily Numbers of ESM Reports and Participants in S1W1.

*Note*: This figure shows how many ESM surveys (blue line) were completed by how many participants (yellow line) per day during the ESM phase of S1W1 (Study 1 Wave 1), that is, from January 14, 2020, until April 17, 2020. ESM report = completed ESM survey. Date format: DD/MM/YYYY. Statistics are based on the respective study wave's state data set.

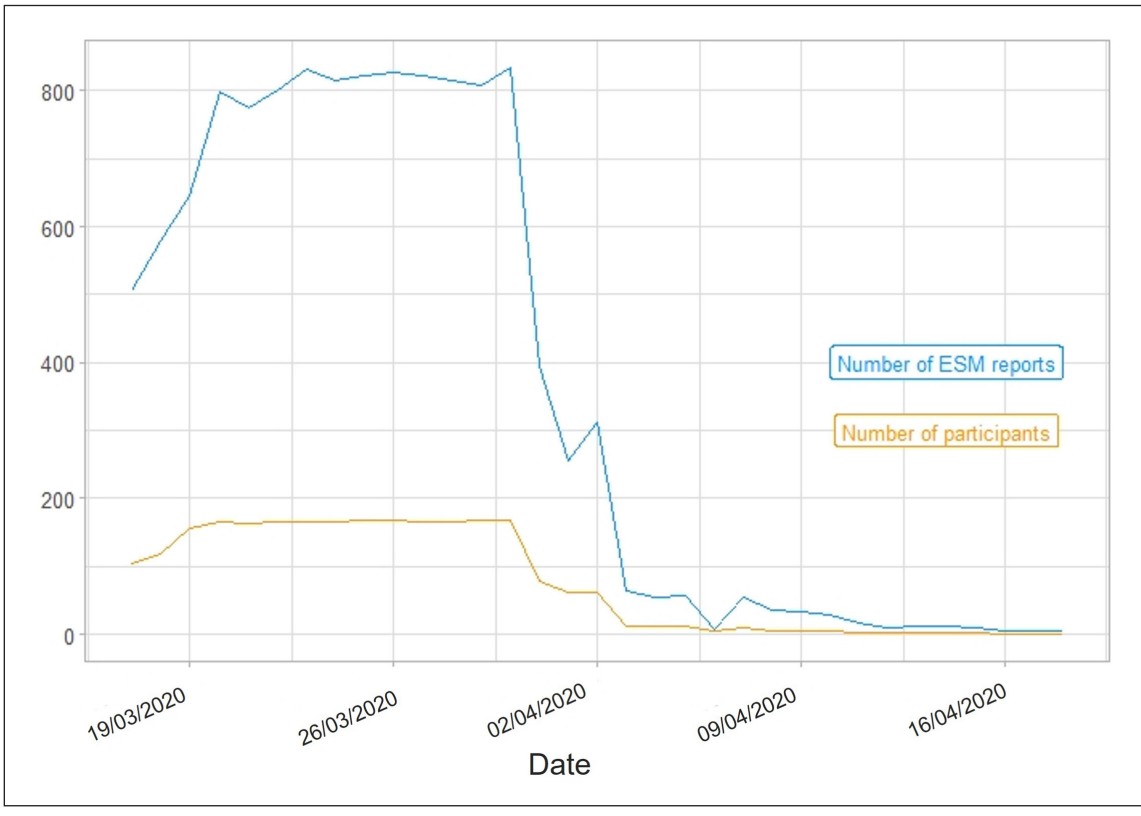

**Figure 3** Longitudinal Trajectories of Daily Numbers of ESM Reports and Participants in S1W2.

*Note*: This figure shows how many ESM surveys (blue line) were completed by how many participants (yellow line) per day during the ESM phase of S1W2 (Study 1 Wave 2), that is, from March 17, 2020, until April 18, 2020. ESM report = completed ESM survey. Date format: DD/MM/YYYY. Statistics are based on the respective study wave's state data set.

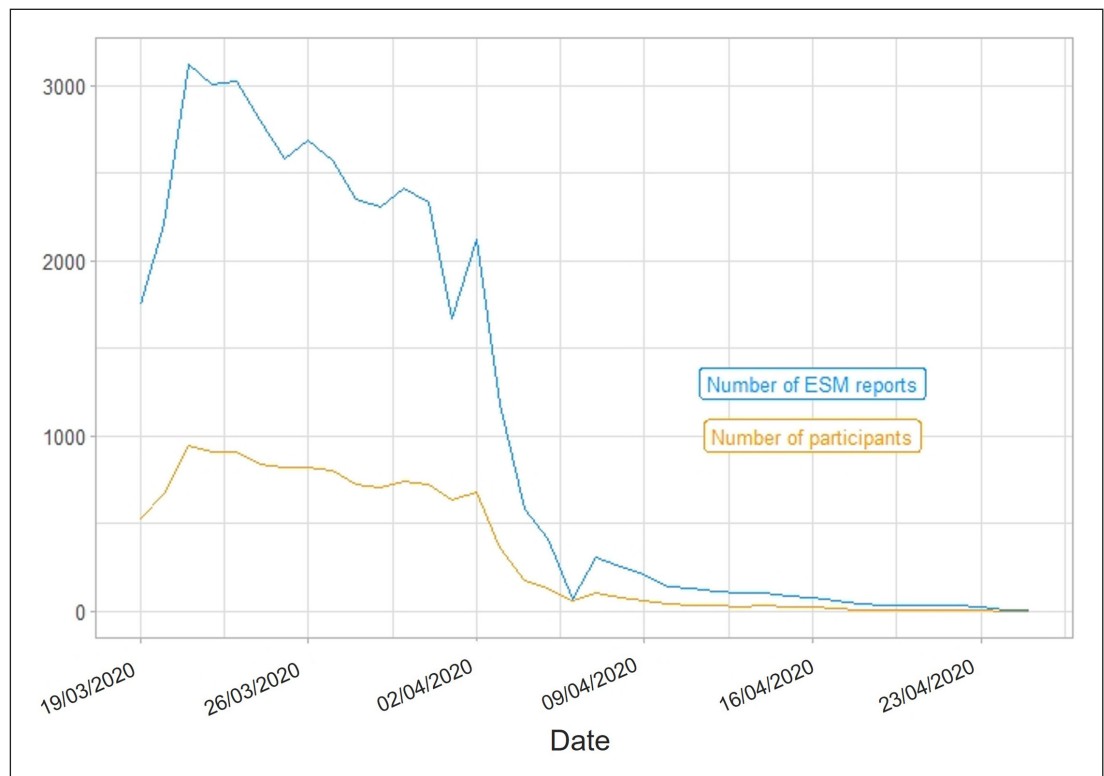

**Figure 4** Longitudinal Trajectories of Daily Numbers of ESM Reports and Participants in S2W1.

*Note*: This figure shows how many ESM surveys (blue line) were completed by how many participants (yellow line) per day during the ESM phase of S2W1 (Study 2 Wave 1), that is, from March 19, 2020, until April 25, 2020. ESM report = completed ESM survey. Date format: DD/MM/YYYY. Statistics are based on the respective study wave's state data set.

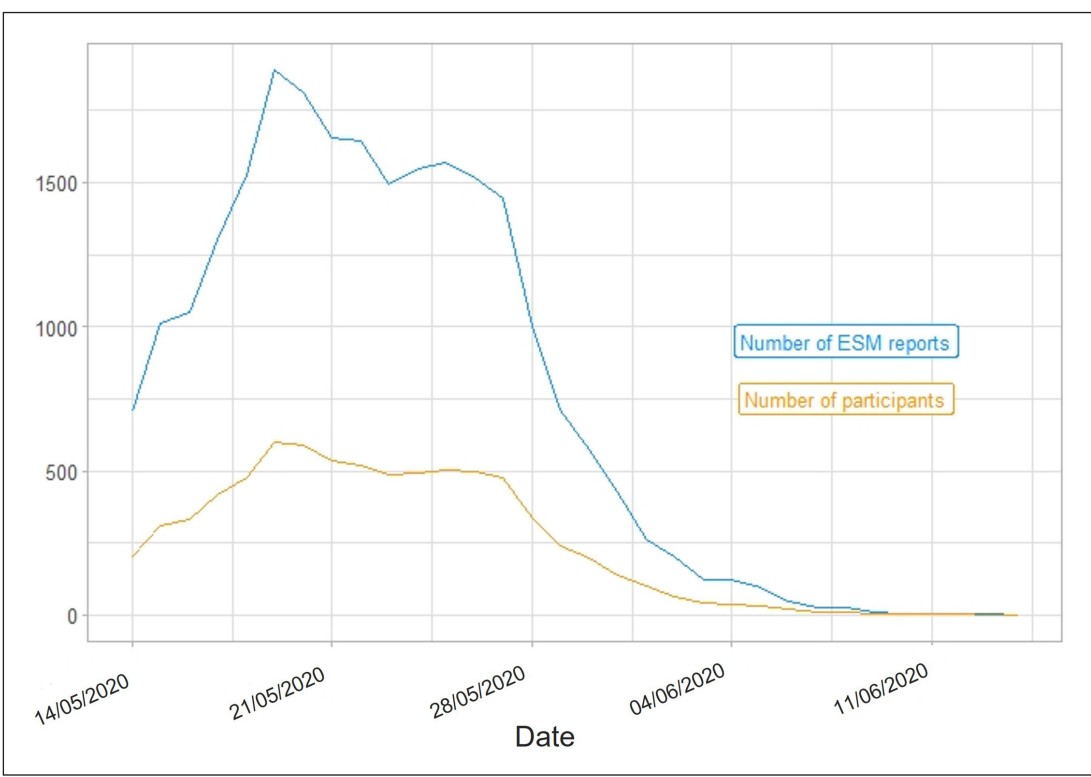

**Figure 5** Longitudinal Trajectories of Daily Numbers of ESM Reports and Participants in S2W2.

*Note*: This figure shows how many ESM surveys (blue line) were completed by how many participants (yellow line) per day during the ESM phase of S2W2 (Study 2 Wave 2), that is, from May 14, 2020, until June 14, 2020. ESM report = completed ESM survey. Date format: DD/MM/YYYY. Statistics are based on the respective study wave's state data set. Please note that no state data was generated beyond June 14, 2020, but that the last possible day to complete the final trait survey was June 16, 2020 (cf. Figure 1 in Section 2.1).

| | NUMBER OF PARTICIPANTS | NUMBER OF STATE REPORTS | ESM REPORTS PER PARTICIPANT / ESM DAYS PER PARTICIPANT | | | |
|---|---|---|---|---|---|---|
| | | | *M* | *MDN* | *SD* | RANGE |
| S1W1 | 313 | 17,464 | 55.8 / 12.1 | 65 / 14 | 25.2 / 4.0 | 1–97 / 1–16 |
| S1W2 | 193 | 12,048 | 62.4 / 12.9 | 71 / 14 | 23.2 / 3.5 | 1–88 / 1–16 |
| S2W1 | 1,645 | 40,994 | 24.9 / 7.8 | 15 / 7 | 24.3 / 5.3 | 1–88 / 1–15 |
| S2W2 | 914 | 23,816 | 26.1 / 8.4 | 19 / 9 | 23.0 / 5.0 | 1–84 / 1–14 |

**Table 3** Descriptive Statistics on ESM Reports and ESM Days per Participant and Study Wave.

*Note*: This table presents means, medians, standard deviations, and ranges pertaining to the number of ESM reports (i.e., completed ESM surveys) and the number of completed ESM days per participant and study wave. S1W1 = Study 1 Wave 1 (all other waves are abbreviated analogously), *Mdn* = median. Statistics are based on the respective study wave's state data set.

| | | STUDY 1 | STUDY 2 |
|---|---|---|---|
| Trait data set | Participants from both study waves incl. Wave 1 and 2 only | $N_{total} = 370$<br>$N_{both} = 186$<br>$N_{S1W1\ only} = 167$<br>$N_{S1W2\ only} = 17$ | $N_{total} = 3{,}565$<br>$N_{both} = 304$<br>$N_{S2W1\ only} = 2{,}211$<br>$N_{S2W2\ only} = 1{,}050$ |
| State data set | Participants from both study waves incl. Wave 1 and 2 only | $N_{participants\ total} = 327$<br>$N_{participants\ both} = 179$<br>$N_{participants\ S1W1\ only} = 134$<br>$N_{participants\ S1W2\ only} = 14$ | $N_{participants\ total} = 2{,}272$<br>$N_{participants\ both} = 287$<br>$N_{participants\ S2W1\ only} = 1{,}358$<br>$N_{participants\ S2W2\ only} = 627$ |
| | State reports from both study waves incl. Wave 1 and 2 only | $N_{state\ reports\ total} = 29{,}512$<br>$N_{state\ reports\ both} = 22{,}903$<br>$N_{state\ reports\ S1W1\ only} = 5{,}749$<br>$N_{state\ reports\ S1W2\ only} = 860$ | $N_{state\ reports\ total} = 64{,}810$<br>$N_{state\ reports\ both} = 22{,}284$<br>$N_{state\ reports\ S2W1\ only} = 27{,}931$<br>$N_{state\ reports\ S2W2\ only} = 14{,}595$ |

**Table 4** Number of Observations in Merged EMOTIONS Data Sets.

*Note*: This table shows the number of participants and—for the state data sets—state reports included in every merged data set. Data sets were merged based on identical emails and/or identical participant codes.

encompassed by every merged data set, decomposed into observations from both study waves, the first study wave only, or the second study wave only.

### 2.4.5 Socio-Demographic Information on the Final Samples

Table 5 presents socio-demographic sample information based on the final trait data set per study wave (with the exception of S1W2, where no socio-demographic variables were assessed). We chose to present socio-demographic information on the trait data sets in the main text because they subsume more participants than the state data sets (see Table 2). The same socio-demographic sample information on each study wave's state data set (again except for S1W2) can be found in Table S1 in the Supplemental Material. Moreover, frequency distributions of age and household size are presented in Figures S1 to S4 in the Supplemental Material.

### 2.4.6 Drop-out-Related Interindividual Differences

To investigate whether or not—and if so, how—individuals who participated in the ESM phase of data collection differed from those who did not, we compared their mean conscientiousness and agreeableness, respectively, using two-sample unpaired *t*-tests.[9] In both waves of Study 1, where the IAL (Jacobs & Scholl, 2005) was administered instead of the BFI-2-S (Rammstedt et al., 2020; for details, see Section 2.5), the LM (warm-agreeable) scale was used as proxy for Agreeableness. Conscientiousness could not be approximated using the IAL. In addition, we explored the associations between the aforementioned personality traits and the number of completed state surveys using bivariate Pearson correlations.[10] Every analysis was performed per study wave (e.g., S1W1). Therefore, we aggregated the appropriate trait scores (i.e., LM in Study 1, Conscientiousness and Agreeableness in Study 2) across the time points at which they were assessed (i.e., T1 and T2 in S1W1 as well as S2W1, T3 and T4 in S2W2, no aggregation but employment of single LM scores at T4 in S1W2).[11]

Results from Study 1 showed that on average, ESM participants were neither significantly[12] more nor less warm-agreeable than those who did not participate in the ESM phase, $M_{ESM} = 4.13$, $SD_{ESM} = 0.50$ vs $M_{no\ ESM} = 3.94$, $SD_{no\ ESM} = 0.62$, $t(333^{[13]}) = -1.69$, $p = .092$, $d = 0.34$ (S1W1).

| SOCIO-DEMOGRAPHIC VARIABLE | STUDY WAVE | | |
|---|---|---|---|
| | **S1W1** | **S2W1** | **S2W2** |
| Gender (% female) | 78.2 | 75.9 | 80.6 |
| Age in years (*M*, *Mdn*, *SD*, range) | 23.0, 21, 6.8, 16–67 | 33.2, 30, 12.6, 16–99 | 41.0, 40, 12.4, 16–75 |
| Educational status (% general qualification for university entrance, % higher education degree) [a] | 80, 18 | 35, 47 | 28, 48 |
| Occupational status (% at university, % currently employed) [b] | 92, 7 | 34, 51 | 11, 69 |
| Current enrollment in higher education (% currently enrolled) [c] | 95 | 37 | 13 |
| Part-time job (% yes) [d] | | 64 | 66 |
| Household size (*M*, *Mdn*, *SD*, range) [e] | | 2.7, 2, 2.7, 1–99 | 2.5, 2, 2.7, 1–90 |
| Relationship status (% single) [f] | | 33 | 28 |

**Table 5** Socio-Demographic Sample Information Based on the Trait Data Set per EMOTIONS Study Wave.

*Note*: This table presents socio-demographic sample information based on the trait data set per EMOTIONS wave. S1W1 = Study 1 Wave 1 (all other waves are abbreviated analogously), *Mdn* = median. Number of participants who provided data on all socio-demographic variables per study wave: $n_{S1W1}$ = 353, $n_{S2W1}$ = 2,515, $n_{S2W2}$ = 1,354. For each socio-demographic sample information, reported statistics are specified in parentheses. Empty cells indicate that the variable in question was not administered in the respective study wave. Note that no socio-demographic information were assessed in S1W2. For full response formats, see our comprehensive codebooks on OSF (osf.io/6kzx3/); and for details on outlier inspection, see Section 2.6.

[a] *General qualification for university entrance* subsumed two response options: 6 (*general qualification for university entrance with no additional vocational training*), 7 (*general qualification for university entrance plus vocational training*). *Higher education degree* subsumed three response options: 8 (*university of applied sciences degree*), 9 (*university degree*), 10 (*university degree and PhD*).

[b] *Currently employed* subsumed three response options: 5 (*full-time employment*), 6 (*part-time employment*), 7 (*self-employed*).

[c] *Currently enrolled* subsumed two response options: 1 (*yes, at a university*), 2 (*yes, at a university of applied sciences*).

[d] Part-time job was assessed from T1 of S2W1 onwards. Moreover, it was displayed only if a participant reported being enrolled in higher education (i.e., at a university or a university of applied sciences), resulting in $n_{S2W1}$ = 916, $n_{S2W2}$ = 178 on this variable.

[e] Household size was assessed from T1 of S2W1 onwards.

[f] Relationship status was assessed from T2 of S2W1 onwards, resulting in $n_{S2W1}$ = 945 on this variable. Moreover, due to drop-out, 1,351 participants provided data on their relationship status in S2W2.

Moreover, being more or less warm-agreeable was not significantly related to the number of completed state surveys, $r(311)$ = .01, $p$ = .860 (S1W1) and $r(168)$ = .12, $p$ = .107 (S1W2).

Results from Study 2 demonstrated that on average, ESM participants were significantly more agreeable than individuals who did not complete any state surveys, $M_{ESM}$ = 3.87, $SD_{ESM}$ = 0.55 vs $M_{no\ ESM}$ = 3.80, $SD_{no\ ESM}$ = 0.54, $t(2356)$ = -2.75, $p$ = .006, $d$ = 0.12 (S2W1).[14] However, no significant group differences in Conscientiousness were revealed, $M_{ESM}$ = 3.70, $SD_{ESM}$ = 0.64 vs $M_{no\ ESM}$ = 3.67, $SD_{no\ ESM}$ = 0.66, $t(2356)$ = -0.93, $p$ = .354, $d$ = 0.04 (S2W1).[15] In contrast, whereas participants' Agreeableness was not significantly associated with the number of completed state surveys, $r(1643)$ = .02, $p$ = .398 (S2W1) and $r(912)$ = .01, $p$ = .850 (S2W2), scores on Conscientiousness were—in S2W1—significantly positively linked to the number of state reports, $r(1643)$ = .07, $p$ = .004.[16] These findings are discussed in Section 4.

## 2.5 MATERIALS

Tables 6 and 7 present all established trait measures employed in the trait surveys of Study 1 and Study 2, respectively. Per measure, each table shows the number of included items, response format, source, as well as mean, standard deviation, reliability (McDonald's omega), and number of participants who provided data on all items per (sub-) scale. For the sake of consistency with the socio-demographic sample information presented in Section 2.4.5, all statistics reported in Tables 6 and 7 are based on the respective study wave's trait data set. The same statistics are provided for each study wave's state data set in Tables S2 and S3 (for Study 1 and Study 2, respectively) in the Supplemental Material.

Table 8 depicts the additional item sets comprising all self-generated, COVID-19-related trait items administered in the trait surveys of the EMOTIONS project. These item sets reflect one possible clustering of items only. Thus, depending on the research question, items from different item sets can be freely combined. Per item set, Table 8 comprises the number of items, a brief summary of what was measured, and the response format. Moreover, it indicates which item sets were assessed per study wave.

Table 9 shows all state items administered in the state surveys of the EMOTIONS project alongside their response formats. The items are organized into self-construed item sets that reflect one possible clustering

| MEASURE, TOTAL NUMBER OF ITEMS, RESPONSE FORMAT, SOURCES (GERMAN, ENGLISH) | SUBSCALE (NUMBER OF ITEMS PER SUBSCALE, IF APPLICABLE) | $M$ (*SD*), ω, $n$ | | |
|---|---|---|---|---|
| | | **S1W1** | | **S1W2** |
| | | **T1** | **T2** | **T4** |
| Interpersonal Adjective Scales (IAL °) + neuroticism-related items<br>• 64 items in the IAL + 16 neuroticism-related items<br>• 1 (*Strongly disagree*); 2 (*Rather disagree*); 3 (*Neutral*); 4 (*Rather agree*); 5 (*Strongly agree*)<br> ◦ Deviation from original response format, that is 1 (*extremely inaccurate*) – 8 (*extremely accurate*)<br>• German version: Jacobs & Scholl (2005)<br>• English version: Wiggins et al. (1988)<br>• Neuroticism-related items were selected from a list by Ostendorf (1994) similar to a selection by Back et al. (2009) | PA (8) | 3.23 (0.63), .82, 335 | 3.31 (0.67), .84, 267 | 3.30 (0.69), .83, 170 |
| | BC (8) | 2.44 (0.67), .79, 336 | 2.40 (0.66), .78, 267 | 2.40 (0.66), .79, 170 |
| | DE (8) | 1.60 (0.50), .78, 335 | 1.56 (0.48), .76, 267 | 1.52 (0.51), .82, 170 |
| | FG (8) | 2.32 (0.66), .82, 335 | 2.26 (0.72), .86, 267 | 2.24 (0.73), .86, 170 |
| | HI (8) | 2.72 (0.73), .81, 335 | 2.67 (0.75), .82, 267 | 2.68 (0.77), .83, 170 |
| | JK (8) | 3.20 (0.54), .72, 335 | 3.21 (0.54), .73, 267 | 3.23 (0.58), .77, 170 |
| | LM (8) | 4.11 (0.53), .82, 335 | 4.14 (0.50), .81, 267 | 4.15 (0.50), .79, 170 |
| | NO (8) | 3.84 (0.54), .81, 335 | 3.88 (0.58), .84, 267 | 3.89 (0.55), .82, 170 |
| | Neuroticism (16) | 2.92 (0.62), .89, 335 | 2.78 (0.62), .89, 267 | 2.73 (0.60), .88, 170 |
| Narcissistic Admiration and Rivalry Questionnaire (NARQ)<br>• 18 items<br>• 1 (*Not agree at all*); 2 (*Not agree*); 3 (*Rather not agree*); 4 (*Rather agree*); 5 (*Agree*); 6 (*Agree completely*)<br> ◦ In the original version, only the extreme poles are labelled verbally, whereas in the EMOTIONS project, all response options were labelled to fulfil the requirements of the matrix response format in formr (Arslan et al., 2020).<br>• German and English version: Back et al. (2013) | Admiration (9) | 3.12 (0.78), .85, 332 | 3.14 (0.80), .87, 266 | 3.14 (0.83), .89, 170 |
| | Rivalry (9) | 2.14 (0.74), .84, 332 | 2.04 (0.74), .85, 266 | 2.06 (0.77), .87, 170 |
| Hypersensitive Narcissism Scale (HSNS)<br>• 10 items<br>• 1 (*Does not apply at all*); 2 (*Rather does not apply*); 3 (*Neutral*); 4 (*Rather applies*); 5 (*Applies completely*)<br> ◦ Deviation from original response format, that is 1 (*very uncharacteristic or untrue, strongly disagree*); 2 (*uncharacteristic*); 3 (*neutral*); 4 (*characteristic*); 5 (*very characteristic or true, strongly agree*)<br>• German version: Morf et al. (2017)<br>• English version: Hendin & Cheek (1997) | | 2.84 (0.49), .49, 328 | 2.85 (0.52), .56, 265 | 2.84 (0.46), .31, 168 |
| Rosenberg Self-Esteem Scale (RSES)<br>• 10 items<br>• 1 (*Strongly disagree*); 2 (*Disagree*); 3 (*Agree*); 4 (*Strongly agree*)<br> ◦ Deviation from original response scale that ranges from 0 to 3<br>• German version: von Collani & Herzberg (2003)<br>• English version: Rosenberg (1965) | | 2.97 (0.52), .89, 332 | 3.03 (0.52), .89, 266 | 3.06 (0.56), .92, 168 |

**Table 6** *Established Trait Measures in EMOTIONS Study 1 (With Statistics Based on Trait Data Sets)*

*Note:* This table shows every established trait measure employed in both waves of Study 1 of the EMOTIONS project, with all descriptive statistics being calculated on the basis of the respective study wave's trait data set. ω = McDonald's omega, $n$ = number of participants who provided data on all items per (sub-) scale, S1W1 = Study 1 Wave 1, S1W2 = Study 1 Wave 2. No established trait measures were assessed at T3 of S1W2. Thus, this time point is omitted from this table. For full sources and the order of assessment, please refer to this paper's reference list and/or the study-wave-specific codebooks. Each codebook offers the most detailed and chronological transcript of every EMOTIONS wave (incl. all instructions, item wordings, and response formats) and can be retrieved from osf.io/6kzx3/.

| MEASURE, TOTAL NUMBER OF ITEMS, RESPONSE FORMAT, SOURCES (GERMAN, ENGLISH) | SUBSCALE (NUMBER OF ITEMS PER SUBSCALE, IF APPLICABLE) | $M\ (SD)$, ω, $n$ | | | |
|---|---|---|---|---|---|
| | | S2W1 | | S2W2 | |
| | | T1 | T2 | T3 | T4 |
| Big Five Inventory-2-S (BFI-2-S)<br>• 30 items<br>• 1 (*Disagree strongly*); 2 (*Disagree a little*); 3 (*Neutral; no opinion*); 4 (*Agree strongly*); 5 (*Agree strongly*)<br>• German version: Rammstedt et al. (2020)<br>• English version: Soto & John (2017) | Negative Emotionality (6) | 2.74 (0.74), .81, 2,358 | 2.67 (0.77), .83, 937 | 2.79 (0.76), .82, 1,299 | 2.72 (0.80), .84, 627 |
| | Extraversion (6) | 3.31 (0.66), .74, 2,358 | 3.21 (0.66), .74, 937 | 3.17 (0.67), .74, 1,299 | 3.12 (0.65), .74, 627 |
| | Open-Mindedness (6) | 3.63 (0.69), .72, 2,358 | 3.68 (0.73), .76, 937 | 3.63 (0.72), .75, 1,299 | 3.71 (0.78), .77, 627 |
| | Agreeableness (6) | 3.85 (0.55), .67, 2,358 | 3.87 (0.57), .71, 937 | 3.79 (0.56), .69, 1,299 | 3.86 (0.54), .67, 627 |
| | Conscientiousness (6) | 3.70 (0.65), .77, 2,358 | 3.69 (0.67), .80, 937 | 3.60 (0.65), .78, 1,299 | 3.62 (0.66), .80, 627 |
| Honesty-Humility (subscale from the HEXACO-60)<br>• 10 items<br>• 1 (*strongly disagree*); 2 (*disagree*); 3 (*neutral*); 4 (*agree*) 5 (*strongly agree*)<br>• German and English version: Ashton & Lee (2009) | | | 3.74 (0.55), .62, 920 | 3.74 (0.54), .59, 1,221 | 3.78 (0.56), .60, 616 |
| Narcissistic Admiration and Rivalry Questionnaire Short Scale (NARQ-S)<br>• 6 items<br>• 1 (*Not agree at all*); 2 (*Not agree*); 3 (*Rather not agree*); 4 (*Rather agree*); 5 (*Agree*); 6 (*Agree completely*)<br> ○ In the original version, only the extreme poles are labelled verbally, whereas in the EMOTIONS project, all response options were labelled to fulfil the requirements of the matrix response format in formr (Arslan et al., 2020).<br>• German and English version: Back et al. (2013)<br>• More recent validation study (of the German and English version): Leckelt et al. (2018) | Admiration (3) | 2.65 (1.03), .78, 2,317 | 2.54 (1.05), .80, 931 | 2.57 (1.04), .78, 1,279 | 2.39 (1.04), .81, 623 |
| | Rivalry (3) | 2.10 (0.81), .64, 2,317 | 2.02 (0.84), .67, 931 | 2.09 (0.82), .64, 1,279 | 1.96 (0.79), .63, 623 |
| Single self-esteem item (from the RSES)<br>• 0 (*Strongly disagree*) – 10 (*Strongly agree*)<br> ○ Deviation from original response format, that is 0 (*Strongly disagree*) – 3 (*Strongly agree*)<br>• German version: von Collani & Herzberg (2003)<br>• English version: Rosenberg (1965) | | 6.83 (1.87), 2,317 | 6.93 (1.84), 931 | 6.60 (2.05), 1,279 | 6.69 (2.02), 623 |

| MEASURE, TOTAL NUMBER OF ITEMS, RESPONSE FORMAT, SOURCES (GERMAN, ENGLISH) | SUBSCALE (NUMBER OF ITEMS PER SUBSCALE, IF APPLICABLE) | $M$ ($SD$), ω, $n$ | | | |
|---|---|---|---|---|---|
| | | S2W1 | | S2W2 | |
| | | T1 | T2 | T3 | T4 |
| UCLA Loneliness Scale (ULS)<br>• 9 items<br>• 1 (*Never*); 2 (*Rarely*); 3 (*Sometimes*); 4 (*Often*); 5 (*Always*)<br> ◦ Deviation from original response format, that is 1 (*never*); 2 (*rarely*); 3 (*sometimes*); 4 (*always*): The response option 4 (*Often*) was added to provide an equally spaced rating scale.<br>• German version: Luhmann et al. (2016)<br>• English version: Russell et al. (1980) | | 2.45 (0.63), 2,281 | 2.47 (0.63), .87, 929 | 2.63 (0.65), .88, 1,262 | 2.57 (0.64), .89, 621 |
| Political orientation<br>• 1 item<br>• 1 (*Left*) – 11 (*Right*)<br>• German version: Kroh (2007)<br>• No English version available | | | 4.33 (1.81), 927 | 4.46 (1.93), 1,245 | 4.33 (1.79), 620 |
| Conspiracy Mentality Questionnaire (CMQ)<br>• 5 items<br>• 1 (*Extremely unlikely*) – 11 (*Extremely likely*)<br> ◦ Deviation from original response format, that is 0% (*certainly not*); 10% (*extremely unlikely*); 20% (*very unlikely*); 30% (*unlikely*); 40% (*somewhat unlikely*); 50% (*undecided*); 60% (*somewhat likely*); 70% (*likely*); 80% (*very likely*); 90% (*extremely likely*); 100% (*certain*)<br>• German and English version: Bruder et al. (2013) | | | 5.19 (2.10), .86, 927 | 5.04 (2.28), .87, 1,245 | 4.66 (2.13), .87, 620 |

**Table 7** Established Trait Measures in EMOTIONS Study 2 (With Statistics Based on Trait Data Sets).

*Note:* This table shows every established trait measure employed in both waves of Study 2 of the EMOTIONS project, with all descriptive statistics being calculated on the basis of the respective study wave's trait data set. ω = McDonald's omega, $n$ = number of participants who provided data on all items per (sub-) scale, S2W1 = Study 2 Wave 1, S2W2 = Study 2 Wave 2. An empty cell indicates that the measure was not assessed at the designated time point. McDonald's omega could not be computed for single-item measures (i.e., single self-esteem item, political orientation). For full sources and the order of assessment, please refer to this paper's reference list and/or the study-wave-specific codebooks. Each codebook offers the most detailed and chronological transcript of every EMOTIONS wave (incl. all instructions, item wordings, and response formats) and can be retrieved from osf.io/6kzx3/.

| MEASURE | NUMBER OF ITEMS | WHAT WAS MEASURED? | RESPONSE FORMAT | S1W2 | | S2W1 | | S2W2 | |
|---|---|---|---|---|---|---|---|---|---|
| | | | | T3 | T4 | T1 | T2 | T3 | T4 |
| Coronavirus-related risk estimations | | | Rating scale: 1 (*Very low*) – 6 (*Very high*) | | | | | | |
| Self-related risk estimations | 3 | Estimated risk of the coronavirus for personal health, social life, and work/university studies | | ✓ | ✓ | ✓ | ✓ | ✓ | ✓ |
| Family-related risk estimations | 3 | Estimated risk of the coronavirus for family's (i.e., parents, grandparents, siblings) health, social life, and working life/university studies | | ✓ | ✓ | ✓ | ✓ | ✓ | ✓ |
| Close-others-related risk estimations | 3 | Estimated risk of the coronavirus for close others' (i.e., partner, close friends) health, social life, and working life/university studies | | ✓ | ✓ | ✓ | ✓ | ✓ | ✓ |
| Society-related risk estimations | 4 | Estimated risk of the coronavirus for the healthcare system, social cohesion, economy/working life, and cultural life | | ✓ | ✓ | ✓ | ✓ | ✓ | ✓ |
| Coronavirus-related worries | | | Rating scale: 1 (*Does not apply at all*) – 6 (*Applies completely*) | | | | | | |
| Personal worries | 3 | Assessment of personal worries due to the coronavirus and its containment (e.g., oneself being anxious/worried) | | ✓ | ✓ | ✓ | ✓ | ✓ | ✓ |
| Others' worries | 3 | Assessment of other people's worries due to the coronavirus and its containment (e.g., others being anxious/worried) | | ✓ | ✓ | ✓ | ✓ | ✓ | ✓ |
| Coronavirus-related behavioral evaluations | | | Rating scale: 1 (*Does not apply at all*) – 6 (*Applies completely*) | | | | | | |
| General behavioral evaluations | 4 | Evaluation of the appropriateness of governmental crisis management, people's panic, journalists' reporting, and the discussion on social media | | ✓ | ✓ | ✓ | ✓ | ✓ | ✓ |
| Self-related behavioral evaluations | 5 | Evaluation of personal behaviors regarding the coronavirus and its containment (e.g., oneself acting with caution) | | ✓ | ✓ | ✓ | ✓ | ✓ | ✓ |
| Personal hope and belief in social cohesion | 2 | Assessment of the beliefs that "we as society can get the situation under control" and that one is "in the same boat as everyone else" | | ✗ | ✗ | ✓ | ✓ | ✓ | ✓ |
| Other-related behavioral evaluations | 5 | Evaluation of other people's behaviors regarding the coronavirus and its containment (e.g., others acting with caution) | | ✓ | ✓ | ✓ | ✓ | ✓ | ✓ |

| MEASURE | NUMBER OF ITEMS | WHAT WAS MEASURED? | RESPONSE FORMAT | S1W2 | | S2W1 | | S2W2 | |
|---|---|---|---|---|---|---|---|---|---|
| | | | | T3 | T4 | T1 | T2 | T3 | T4 |
| Coronavirus-related behavioral changes | | Assessment of engaging less, equally, or more often than before the coronavirus crisis in behaviors including: | Rating scale: 1 (*Significantly less often*) – 4 (*Equally often*) – 7 (*Significantly more often*) | | | | | | |
| Hygiene-related behavioral changes | 2 | Washing hands, sneezing/coughing into the crook of one's arm | | ✓ | ✓ | ✓ | ✓ | ✓ | ✓ |
| Social-distancing-related behavioral changes | 4 | Keeping distance to other people, grocery shopping in supermarkets, visiting public places (e.g., markets, shopping centers), staying at home | | ✓ | ✓ | ✓ | ✓ | ✓ | ✓ |
| Travel- and transportation-related behavioral changes | 3 | Going on planned trips, riding one's bike, using public transport | | ✓ | ✓ | ✓ | ✓ | ✓ | ✓ |
| Leisure-related behavioral changes | 13 | Visiting family, meeting up with friends, attending public events with more/less than 50 persons, attending private meetings with more/less than 10 persons, inviting people to one's home, going to nightclubs/cafes, visiting cultural facilities (e.g., museums, cinemas), going to a fitness studio/for a walk/jogging outside | | ✓ | ✓ | ✓ | ✓ | ✓ | ✓ |
| Work-related behavioral changes | 3 | Working/studying at home, studying in the library, sitting for exams | | ✓ | ✓ | ✓ | ✓ | ✓ | ✓ |
| Coronavirus-related stockpiling | 2 | Number of toilet paper rolls and packages of pasta currently in one's household | Integer | ✗ | ✗ | ✓ | ✓ | ✓ | ✓ |
| Coronavirus-related policy evaluations | | Evaluation of the appropriateness of policies regarding the … | Rating scale: 1 (*Not sensible at all*) – 6 (*Extremely sensible*) | | | | | | |
| Evaluation of policies regarding events | 3 | …cancellation of different-sized events (e.g., large-scale events, private events with more than 50 persons) | | ✓ | ✓ | ✓ | ✓ | ✓ | ✓ |
| Evaluation of policies regarding travel as well as (local and long-distance) public transport | 7 | …discontinuation of public transport (e.g., local public transport service, short- and long-haul air travel) and travel bans (e.g., border closures for people seeking to enter the country, ban on travelling abroad) | | ✓ | ✓ | ✓ | ✓ | ✓ | ✓ |

(Contd.)

| MEASURE | NUMBER OF ITEMS | WHAT WAS MEASURED? | RESPONSE FORMAT | S1W2 | | S2W1 | | S2W2 | |
|---|---|---|---|---|---|---|---|---|---|
| | | | | T3 | T4 | T1 | T2 | T3 | T4 |
| Evaluation of policies regarding educational institutions | 3 | …closure of (nursery) schools and higher education institutions | | ✓ | ✓ | ✓ | ✓ | ✓ | ✓ |
| Evaluation of policies regarding cultural life | 3 | …closure of cultural institutions including nightclubs, bars/cafés, museums, or sports facilities | | ✓ | ✓ | ✓ | ✓ | ✓ | ✓ |
| Evaluation of policies regarding retail | 1 | …of retail stores with the exception of supermarkets, drugstores, and pharmacies | | ✓ | ✓ | ✓ | ✓ | ✓ | ✓ |
| Evaluation of policies regarding quarantine and rationing of essential goods | 3 | …imposition of quarantine and rationing/seizure-of-essential-goods regulations | | ✓ | ✓ | ✓ | ✓ | ✓ | ✓ |
| Evaluation of policies regarding the relaxation of restrictions | 1 | Evaluation whether restrictions are being relaxed with appropriate speed | Rating scale: 1 (*Way too slowly*) – 7 (*Way too fast*) | ✗ | ✗ | ✗ | ✗ | ✓ | ✓ |
| **Exposure to the coronavirus** | | | | | | | | | |
| Self-related exposure | 5 in S1W2, 8 starting with S2W1 T1 | Personal exposure to the coronavirus in terms of being in (voluntary or prescribed) quarantine, having/having had typical symptoms of the disease, and having been tested (positive) *Additionally since S2W1 T1:* Personal exposure in terms of following social distancing measures, thinking that one or more people from one's household belong to risk group, and different degrees of being able to pursue one's occupation/part-time job | Select one: 1 (*Yes*); 2 (*No*) | ✓ | ✓ | ✓ | ✓ | ✓ | ✓ |
| Family-related exposure | 5 in S1W2, 3 starting with S2W1 T1 | Number of people in one's family (i.e., parents, grandparents, siblings) being in (voluntary or prescribed) quarantine, reporting symptoms, and having been tested (positive) *Since S2W1 T1:* Only number of people in one's family being in quarantine (no differentiation between prescribed and voluntary quarantine), and having been tested (positive) | Integer | ✓ | ✓ | ✓ | ✓ | ✓ | ✓ |

(Contd.)

| MEASURE | NUMBER OF ITEMS | WHAT WAS MEASURED? | RESPONSE FORMAT | S1W2 | | S2W1 | | S2W2 | | |
|---|---|---|---|---|---|---|---|---|---|---|
| | | | | T3 | T4 | T1 | T2 | T3 | T4 | |
| Close-others-related exposure | 5 in S1W2, 3 starting with S2W1 T1 | Number of people in one's close personal environment (i.e., partner, close friends) being in (voluntary or prescribed) quarantine, reporting symptoms, and having been tested (positive) *Since S2W1 T1*: Only number of people in one's close personal environment being in quarantine (no differentiation between prescribed and voluntary quarantine), and having been tested (positive) | Integer | ✓ | ✓ | ✓ | ✓ | ✓ | ✓ | |
| Wider-social-environment-related exposure | 5 in S1W2, 3 starting with S2W1 T1 | Number of people in one's wider social environment (i.e., fellow university students, other acquaintances) being in quarantine, reporting symptoms, and having been tested (positive) *Since S2W1 T1*: Only number of people in one's wider social environment being in quarantine (no differentiation between prescribed and voluntary quarantine), and having been tested (positive) | Integer | ✓ | ✓ | ✓ | ✓ | ✓ | ✓ | |
| Coronavirus-related personal restrictions | 4 | Assessment of the degree to which one is willing to accept personal quality of life constraints to reduce the risk of infection for oneself, one's family members/friends, others in general, and members of the risk group | Rating scale: 1 (*Disagree strongly*) – 5 (*Agree strongly*) | ✗ | ✗ | ✗ | ✗ | ✓ | ✓ | |

**Table 8** Self-Generated, COVID-19-Related Trait Measures of the EMOTIONS Project.

*Note:* This table presents all self-generated, COVID-19-related trait measures employed in the EMOTIONS project, organized into self-construed item sets. Each item set is accompanied by the number of (self-generated) items it subsumes, an overview of what it measured, and its response format. In addition, a check mark (✓) indicates that the item set was assessed at the designated time point, whereas a cross (✗) indicates that the item set was not assessed at the designated time point. No self-generated, COVID-19-related trait items were administered in S1W1. Thus, this wave is omitted from this table. Number of participants who provided data on all self-generated, COVID-19-related trait measures per study wave and time point in the trait data set (number of participants in the respective state data set given in parentheses): $n_{S1W2\ T3}$ = 201 (193), $n_{S1W2\ T4}$ = 170 (170), $n_{S2W1\ T1}$ = 2,097 (1,645), $n_{S2W1\ T2}$ = 945 (942), $n_{S2W1\ T3}$ = 1,132 (914), $n_{S2W2\ T4}$ = 634 (629). For item-specific information, instructions, and the order of assessment, please refer to the study-wave-specific codebooks. Each codebook offers the most detailed and chronological transcript of every EMOTIONS wave (incl. all instructions, item wordings, and response formats) and can be retrieved from osf.io/6kzx3/.

| MEASURE | INSTRUCTION AND ITEM | RESPONSE FORMAT |
|---|---|---|
| Interaction occurred? | Since the last survey, I (at least) had one social interaction that lasted longer than 5 minutes. | Select one: 1 (*Yes*); 2 (*No*) |
| If interaction | | |
| Type of activity | During what type of activity did the interaction take place? | Select one: 1 (*job-related task/chore*); 2 (*private task/chore*); 3 (*leisure activity*) |
| Mode of communication | The interaction evaluated here took place as follows: | Select one: 1 (*directly/in person*); 2 (*via phone/chat*) |
| Number of interaction partners | How many people other than you were involved in the interaction? Please enter 5 if you interacted with more than 5 people. In this case, please refer below to the five people with whom you interacted the most. | Select one: 1 – 5 |
| Relationship to each interaction partner | Now, for each of the involved interaction partners, please indicate what role he/she had in relation to you. If you interacted with more than 5 people, please report on the 5 most important people in the interaction.<br>This interaction partner in the situation has the following relationship to me: | Select one: 1 (*Supervisor*); 2 (*My employee*); 3 (*Co-worker*); 4 (*Customer client patient*); 5 (*Friend/acquaintance*); 6 (*Partner*); 7 (*My child*); 8 (*Parent*); 9 (*Sibling*); 10 (*other relatives*); 11 (*other persons*) |
| Behavioral states | During the interaction, I exhibited the following behavior: | Rating scale: 1 (*Does not apply at all*) – 6 (*Applies completely*) |
| | I took the lead. | |
| | I criticized others. | |
| | I did not get involved. | |
| | I was self-assured. | |
| | I was unfriendly. | |
| | I was reserved. | |
| | I raised the topic of the coronavirus.[†] | |
| | I helped others.[†] | |
| Perceptual states | During the interaction, I perceived the following: | Rating scale: 1 (*Does not apply at all*) – 6 (*Applies completely*) |
| | I was admired. | |
| | I was criticized. | |
| | I was ignored. | |
| | I was respected. | |
| | Others tried to steal the show from me. | |
| | I was sidelined. | |
| | I was asked about the coronavirus.[†] | |
| | I experienced understanding and a feeling of security from others.[†] | |
| Emotional states | How did you feel immediately after the interaction? | Rating scale: 1 (*Does not apply at all*) – 6 (*Applies completely*) |
| | Proud | |
| | Successful | |
| | Superior | |

(Contd.)

| MEASURE | INSTRUCTION AND ITEM | RESPONSE FORMAT |
|---|---|---|
| | Angry | |
| | Socially excluded† | |
| | Envious | |
| | Resentful | |
| | Ashamed | |
| | Insecure | |
| | Enthusiastic | |
| | Relaxed | |
| | Anxious | |
| | Sad | |
| | Lonely† | |
| | Finally, please use the following sliders to indicate how dissatisfied vs satisfied and calm vs activated you felt overall: | Select one using slider: 0 – 100 |
| | dissatisfied vs satisfied | |
| | calm vs activated | |
| If no interaction (i.e., if non-social activity) | | |
| Type of activity | What kind of activity was it? | Select one: 1 (*job-related task/chore*); 2 (*private task/chore*); 3 (*leisure activity*) |
| Mode of activity | The activity evaluated here took place as follows: | Select one: 1 (*on the computer/laptop/tablet/cell phone*); 2 (*not on the computer/laptop/tablet/cell phone*) |
| COVID-19-specific activities | The activity related to: | Select one: 1 (*Yes*); 2 (*No*) |
| | Researching the coronavirus† | |
| | Reading news about the coronavirus† | |
| Perceptual states | During the activity, I perceived the following: | Rating scale: 1 (*Does not apply at all*) – 6 (*Applies completely*) |
| | I found the activity pleasant. | |
| | I had fun. | |
| | I did tasks that others assigned to me. | |
| | I was intellectually/mentally stimulated. | |
| | I was overwhelmed. | |
| | I was bored. | |
| | I was concentrated. | |
| | I was motivated. | |
| | I thought about the coronavirus.† | |
| Emotional states | How did you feel immediately after the activity? | Rating scale: 1 (*Does not apply at all*) – 6 (*Applies completely*) |
| | Proud | |

| MEASURE | INSTRUCTION AND ITEM | RESPONSE FORMAT |
|---|---|---|
| | Successful | |
| | Superior | |
| | Angry | |
| | Socially excluded[†] | |
| | Envious | |
| | Resentful | |
| | Ashamed | |
| | Insecure | |
| | Enthusiastic | |
| | Relaxed | |
| | Anxious | |
| | Sad | |
| | Lonely[†] | |
| | Finally, please use the following sliders to indicate how dissatisfied vs satisfied and calm vs activated you felt overall: | Select one using slider: 0 – 100 |
| | dissatisfied vs satisfied | |
| | calm vs activated | |
| Interaction-independent, coronavirus-related momentary worries | Due to the coronavirus outbreak, I am worried about … | Rating scale: 1 (*Very little*) – 6 (*Very much*) |
| Self-related worries | … my health.* | |
| | … my social life.* | |
| | … my university studies/my work.* | |
| Society-related worries | … the healthcare system in Germany.* | |
| | … social cohesion in Germany.* | |
| | … the economy/working life in Germany.* | |
| | … cultural life in Germany.* | |
| Other-related worries (voluntary items: if an item did not apply, it could be skipped) | | |
| Parents-related worries | … my parents' health.* | |
| Grandparents-related worries | … my grandparents' health.* | |
| Siblings-related worries | … my siblings' health.* | |
| Children-related worries | … my children's health.[†] | |
| Partner-related worries | … my partner's health.* | |
| Close-friends-related worries | … my close friends' health.* | |

| MEASURE | INSTRUCTION AND ITEM | RESPONSE FORMAT |
|---|---|---|
| Wider-social-environment-related worries | … the health of my wider social environment (fellow university students, other acquaintances).* | |

**Table 9** State Measures of the EMOTIONS Project.

*Note*: This table presents all state items administered in the EMOTIONS project, organized into self-construed item sets and accompanied by each item's response format. Some items were adapted from the Interpersonal Adjective Scales (IAL; German version by Jacobs & Scholl, 2005; English version by Wiggins et al., 1988), the Positive and Negative Affect Schedule (PANAS; German versions by Krohne et al., 1996; Röcke & Grühn, 2003; English version by Watson et al., 1988), and the affect grid (Russell et al., 1989). Other items and all response formats were self-generated. Items marked with an asterisk (*) were administered from S1W2 onwards, and items marked with a cross (†) were administered from S2W1 onwards. Number of participants who and state reports that provided data on all state items (except for those items that assessed [a] the relationship to each interaction partner due to their dependency on the number of interaction partners and [b] interaction-independent, coronavirus-related momentary worries due to their mostly voluntary nature) per study wave (either post social interaction or non-social activity): $n_{S1W1\ participants}$ = 313 (interaction = 238; activity = 75), $n_{S1W1\ ESM\ reports}$ = 17,464 (interaction = 10,548; activity = 6,916), $n_{S1W2\ participants}$ = 193 (interaction = 146; activity = 47), $n_{S1W2\ ESM\ reports}$ = 12,048 (interaction = 7,486; activity = 4,562), $n_{S2W1\ participants}$ = 1,645 (interaction = 1,152; activity = 493), $n_{S2W1\ ESM\ reports}$ = 40,994 (interaction = 25,210; activity = 15,784), $n_{S2W2\ participants}$ = 914 (interaction = 698; activity = 216), $n_{S2W2\ ESM\ reports}$ = 23,816 (interaction = 14,663; activity = 9,153). For full instructions and the order of assessment, please refer to the study-wave-specific codebooks. Each codebook offers the most detailed and chronological transcript of every EMOTIONS wave (incl. all instructions, item wordings, and response formats) and can be retrieved from: osf.io/6kzx3/.

[a] Participants were interrogated on their relationships with—for instance—a third interaction partner only if they reported that three or more people were involved in the preceding interaction.

of items only and are open to modification. Depending on whether participants experienced a social interaction or non-social activity prior to survey completion, they responded to two partially different blocks of items. Thus, Table 9 (just as the EMOTIONS codebooks; see osf. io/6kzx3/) distinguishes between momentary states assessed post social interaction vs post non-social activity.

## 2.6 QUALITY CONTROL

In addition to the data exclusion procedures detailed in Section 2.4.2, we conducted data quality checks separately on the trait and state data sets per EMOTIONS wave. Therein, we calculated various data quality metrics first, pursuing a participant- and a variable-oriented approach. Data quality metrics pertaining to the participant-oriented approach (i.e., intraindividual response variability, longstring, survey completion time) quantified conspicuous responding within individual participants and across a set of variables. The data quality metric pertaining to the variable-oriented approach quantified conspicuous responses on all metric variables with open answer format (e.g., age, household size). Next, per data quality metric, participants with outlying values were flagged on newly created dichotomous variables. Finally, both approaches were integrated to identify participants with at least one outlying value (for details on this procedure, see below). Note that we did not delete any data during the data quality checks. Instead, all data quality metrics and variables indicating flagged participants were added to the respective data set. Thereby, researchers can filter the data, inspect the specific responses and/ or response patterns of flagged participants, and make

an informed decision upon further data exclusion (or retention).

In the following, we introduce our criteria for conspicuous responding on each (participant- and variable-oriented) data quality metric. For a summary of all data quality checks that were executed per study wave and time point, please refer to Table S4 in the Supplemental Material.

### 2.6.1 Participant-Oriented Quality Control
#### 2.6.1.1 Trait Measures
We examined three metrics of careless responding to inspect data quality per participant: (a) intraindividual response variability (IRV), (b) longstring, and (c) survey completion time. The first two metrics (i.e., IRV and longstring) were calculated separately per established measure (e.g., IAL), study wave (e.g., S1W1), and time point (e.g., T1). The latter metric (i.e., completion time) was calculated per trait survey (i.e., per study wave and time point). Participants with an outlying IRV, longstring, or completion time were flagged by creating dichotomous variables, which took on a value of 1 for potential[17] outliers and 0 otherwise (e.g., *outlier_irv_ial_t1* for outlying IRVs on the IAL at T1 of S1W1).

**(a)** IRV is quantified by the standard deviation of a single participant's responses on "a set of consecutive items" (Dunn et al., 2018, p. 108). It can be computed across all or a (sufficiently long) subset of items a participant responded to, as long as these items share a common response format. Since we administered measures with different response formats in the EMOTIONS trait surveys (for details, see Section 2.5), we calculated

IRV separately for different measures. Therein, we only considered measures that were at least 10 items long and established (i.e., psychometrically validated).[18] An outlying IRV was (i) unusually low (i.e., below 25%-percentile - 3 × IQR[19]; see recommendation by Dunn et al., 2018) for measures subsuming multiple constructs and/or including reverse-coded items (i.e., IAL, BFI-2-S, RSES, and Honesty-Humility[20]) and (ii) unusually high (above 75%-percentile + 3 × IQR; see recommendation by Marjanovic et al., 2015) for measures subsuming a single construct and/or not including reverse-coded items (i.e., NARQ and HSNS).

**(b)** A longstring corresponds to the longest consecutive string of identical responses (Dunn et al., 2018). For instance, a longstring of 17 indicates that the participant chose the same response on 17 consecutive items. The longstring metric was calculated for established scales with 10 or more items that measure multiple constructs and/or include reverse-coded items only (i.e., IAL, RSES, BFI-2-S, Honesty-Humility). In contrast, on scales that measure a single construct or where all items are coded in the same direction (i.e., NARQ, HSNS), long longstrings can be appropriate representations of participants' self-reported trait expressions (e.g., a person who reports being very non-narcissistic could hypothetically choose response option 1 across all items of the NARQ). Thus, we refrained from inspecting the longstring metric on the NARQ and HSNS. Longstring values greater than the 75%-percentile + 3 × IQR were considered outliers.

**(c)** Careless responding can also resonate in conspicuously short survey completion times (Dunn et al., 2018; Meade & Craig, 2012). Specifically, if a participant rushed through a survey, a careful read of instructions and items was unlikely or even impossible. We stipulated 2 s as the minimum duration necessary to read and respond to an item carefully. Thus, survey completion times[21] that were shorter than 2 s × number of completed items per survey were identified as outliers on the completion time metric and flagged using the aforementioned procedure.

### 2.6.1.2 State Measures

In the state surveys, we employed the previously described data quality metrics (i.e., IRV, longstring, completion time) to identify conspicuous responding. First, we inspected each data quality metric per measurement occasion (i.e., state report). Then, per metric, we created an overall dichotomous variable that marked participants with at least one measurement-occasion-specific outlier.

**(a)** Although no full established scale was assessed in the state surveys, per measurement occasion,

we computed IRV across all social-interaction- or activity-related state items with the same response format. This is due to the inclusion of items reflecting multiple constructs from established scales like the IAL or PANAS, in which case identical responding across an entire state survey may reasonably be considered conspicuous. Per measurement occasion, IRVs below the 25%-percentile - 3 × IQR were considered outliers.

**(b)** Analogously, the longstring metric was inspected per measurement occasion across all social-interaction- or activity-related state items with the same response format. Longstring values greater than the 75%-percentile + 3 × IQR were considered outliers.

**(c)** State surveys that ended after less than 1 s[22] × number of completed, social-interaction- or activity-related state items were considered outliers.

### 2.6.2 Variable-Oriented Quality Control

Besides inspecting data quality per participant, we scrutinized variables with open answer format for conspicuous answers. This check was performed on trait measures only, because no variables with open answer format were assessed in the state surveys. We distinguished between metric and qualitative variables with open answer format. Regarding the former (e.g., age, household size, number of family members in quarantine in S2W1), values above a threshold of 75%-percentile + 3 × IQR were identified as outliers and flagged by assigning the value 1 to a newly created dichotomous variable (e.g., *outlier_household_t1* for T1 of S2W1). In contrast, no dichotomous variable was created for qualitative variables with open answer format (e.g., field of university studies), since their outliers could not be detected by algorithmic means. Instead, a descriptive comment was made in the respective data preparation R script.

### 2.6.3 Integrating Participant- and Variable-Oriented Quality Controls

Finally, to facilitate the identification of participants with some conspicuous responding (i.e., some outlying data quality metric), we created an overall dichotomous variable per EMOTIONS study wave (e.g., *outlier_participant_s1w1*). Following the previously-outlined logic, this variable was assigned the value 1 if a participant was flagged on at least one of the aforementioned, participant- or variable-oriented data quality metrics.

## 2.7 DATA ANONYMIZATION AND ETHICAL ISSUES

The procedures of all EMOTIONS waves were approved by the review board of the University of Münster, Germany. To anonymize the data, four steps were taken. First, the data were collected pseudonymized. That is, participants

created a personal participant code by concatenating the first letter of their mother's first name (e.g., Karin = K), their mother's month of birth as a number (e.g., May = 05), the last letter of their first name (e.g., Steve = E), their month of birth as a number (e.g., February = 02), and the last letter of their surname (e.g., Smith = H) to create an order of letters and numbers (here: K05E02H). The codes were needed to (a) merge data across waves per study (for more information on the merged data sets, see Section 2.4.4), (b) issue course credits for compensation after data collection (in Study 1 only; see also Section 2.4.1.2), and (c) enable the exclusion of data. Second, personal and identifiable email addresses that were specified to participate in each study wave's ESM phase were anonymized post data collection by assigning a random number to each email address. Third, in the course of data processing, every unique participant was assigned a participant ID—a number between 1 and $N$ (i.e., the sample size in the respective data set) that anonymized the data. And fourth, from all data sets that were published on osf. io/6kzx3/, we removed variables that could facilitate the identification of participants, namely the participant code, gender, gender specification (i.e., a variable with open answer format enabling the specification of a non-binary gender), age, semester of studies, and field of studies.[23]

## 2.8 EXISTING USE OF DATA

To date, there are two published papers that used EMOTIONS data. The first paper by Kroencke et al. (2020) investigated the association between neuroticism and negative affect, considering the latter's longitudinal trajectory over the course of the 14-day ESM phase. The authors used trait and state data from S2W1. The second paper was an international collaboration by Zettler et al. (2022) that explored the links between multiple personality traits and COVID-19-related responses. The researchers combined data from Denmark with trait data from the first and second wave of Study 2. In addition, there are further papers in preparation (e.g., Kroencke, Humberg, et al., 2022; Kroencke, Kuper et al., 2022). On the EMOTIONS project page at osf.io/6kzx3/, we provide a complete and continuously updated list of all published as well as forthcoming manuscripts.

## 3. DATA SET DESCRIPTION AND ACCESS

Table 10 presents all details pertaining to the EMOTIONS data sets that we share collectively on osf.io/6kzx3/.

## 4. REUSE POTENTIAL

We conducted two large-scale (Study 1: $N_{ESM\ participants}$ = 327 and $N_{ESM\ reports}$ = 29,512, Study 2: $N_{ESM\ participants}$ = 2,272

and $N_{ESM\ reports}$ = 64,810), multi-wave ESM studies among different samples (i.e., university students in Study 1 and members of the general population in Study 2). Thereby, we contribute to the methodological variety of psychological COVID-19 research, which—to date—has been dominated by cross-sectional studies employing global measures (Sterl et al., 2022; for two illustrative examples, see Lazarević et al., 2021; Peitz et al., 2021).

ESM enabled us to generate fine-grained and complex, longitudinal data on the frequency, intensity, and variability of psychological states as well as their situational context (e.g., Larson & Csikszentmihalyi, 1983; Scollon et al., 2009). These data permit manifold analyses on the within- and between-person level. On the within-person level, researchers can explore the intraindividual dynamics of momentary states (e.g., intraindividual variability, time trajectories, situation-state contingencies; Beal, 2015; Funder, 2015; Scollon et al., 2009). On the between-person level, investigators can scrutinize interindividual differences in mean state levels and intraindividual dynamics. And since we collected data across multiple study waves, the stability of and change in trait measures as well as (ESM-based) state measures (i.e., mean state levels and intraindividual dynamics, respectively) can be examined on the within-person level.

ESM helped us to avoid some pitfalls associated with decontextualized or retrospective self-reports, such as memory biases or heuristic evaluations (e.g., Larson & Csikszentmihalyi, 1983; Raphael, 1987). We achieved this goal by keeping the time lag between prompt and response short (45 min at the most) and explicitly asking participants to recollect their last social interaction or non-social activity.

Moreover, we acquired data on comparatively large samples (especially in Study 2). This is a noteworthy deviation from common practice in ESM research, where relatively small samples were found to constitute the norm (Wrzus & Mehl, 2015; cf. Xu et al., 2021). Consequently, statistical analyses based on EMOTIONS data can be equipped with greater precision and power, whereby the latter may facilitate detecting effect sizes of varying—and especially smaller—magnitude (Eid et al., 2017). Notably, in the context of COVID-19, even small effects can make a big difference (Funder & Ozer, 2019), if they imply that health-prevention behaviors might be hampered on the societal level or that spread-mitigating efforts could be undermined on the global scale (Bierwiaczonek et al., 2022).

Three possible limitations to our data and data collection process must be recognized. The first limitation pertains to the ESM component of the EMOTIONS project and concerns situation and participant (self-) selection. Regarding situation selection, given that participants could freely choose when (not) to complete a state survey, responses in more unusual or extreme situations (e.g.,

|  |  | STUDY 1 | | STUDY 2 | |
|---|---|---|---|---|---|
|  |  | WAVE 1 | WAVE 2 | WAVE 1 | WAVE 2 |
| File name and data type | Trait data (processed data) | Study1_Wave1_Traitdata.csv | Study1_Wave2_Traitdata.csv | Study2_Wave1_Traitdata.csv | Study2_Wave2_Traitdata.csv |
|  | Merged trait data (merged from the processed data sets above) | Study1_BothWaves_Traitdata_inclWave1[2]-only.csv | | Study2_BothWaves_Traitdata_inclWave1[2]-only.csv | |
|  | ESM data (processed data) | Study1_Wave1_ESMdata.csv | Study1_Wave2_ESMdata.csv | Study2_Wave1_ESMdata.csv | Study2_Wave2_ESMdata.csv |
|  | Merged ESM data (merged from the processed data sets above) | Study1_BothWaves_ESMdata_inclWave1[2]-only.csv | | Study2_BothWaves_ESMdata_inclWave1[2]-only.csv | |
| Format name and version | | CSV All state (i.e., ESM) data sets are in long format. | | | |
| Language | | American English | | | |
| License | | CC-By Attribution 4.0 International | | | |
| Limits to sharing | | All EMOTIONS data are shared on osf.io/6kzx3/. We encourage researchers wishing to use (subsets of the EMOTIONS data sets) to link their preregistrations on OSF with the EMOTIONS project OSF page. Preregistrations can be created directly via OSF Registries (osf.io/registries/osf/new). Likewise, any other approach to preregistration is welcome (e.g., a self-generated preregistration file that is uploaded on OSF). In this preregistration, we ask researchers to specify their research objective(s)/question(s), hypotheses, and the EMOTIONS data to be used. This procedure will allow other investigators and us to keep an overview of all planned and ongoing research projects that employ EMOTIONS data, minimizing potential overlap between different research projects. Moreover, we created a Google Survey (https://forms.gle/MDj6WceMcioq5eUt9), where we kindly ask all researchers planning to utilize the EMOTIONS data to provide brief information on themselves, their research question(s), and the EMOTIONS data of interest. By providing such up-to-date, easily-generated information on upcoming research projects, this survey is supposed to supplement the more formal preregistration mentioned above. As noted in Section 2.7, we took several measures to anonymize the data prior to its publication. One measure was the removal of possibly identifiable variables (participant code, gender, gender specification, age, semester of studies, and field of studies). However, all variables (except for participant code) can be personally requested from us. | | | |
| Publication date | | 04/11/2022 | | | |
| FAIR data/ Codebooks | | Codebook_EMOTIONS_Study1_Wave1.pdf, Codebook_EMOTIONS_Study1_Wave2.pdf, Codebook_EMOTIONS_Study2_Wave1.pdf, and Codebook_EMOTIONS_Study2_Wave2.pdf to be retrieved from OSF | | | |

**Table 10** Description of all EMOTIONS Data Sets.

*Note*: This table provides detailed information on all shared EMOTIONS data sets. S1W1 = Study 1 Wave 1 (all other waves are abbreviated analogously). "_inclWave1[2]-only" indicates that participants who completed either S1W1 or S1W2 (likewise: S2W1 or S2W2) only have been included in addition to participants who completed both waves.

interpersonal conflicts; Schimmack, 2003) may be less likely. This might limit the "degree to which a full range of participants' activities and situations are sampled" (Scollon et al., 2009, p. 17) and could diminish the ecological validity of findings. Concerning participant self-selection, individuals who initially chose and—in the course of data collection—continued to complete the state surveys may differ from others in systematic ways (e.g., greater motivation, conscientiousness, agreeableness; Scollon et al., 2009). In fact, in Study 2, ESM participants self-reported being more agreeable than individuals who dropped out prior to the ESM phase of data collection; and in S2W1, higher levels of Conscientiousness were significantly positively associated with the number of completed state surveys (for details, see Section 2.4.6). However, effect sizes were consistently small (with Pearson correlations never exceeding .12 and Cohen's d never surpassing an absolute value of 0.34), and statistical significance was likely induced by large sample sizes in Study 2[24] (Funder & Ozer, 2019). Thus, the aforementioned insights ought to be interpreted with caution, their inconclusive nature[25] should be considered, and additional possible intergroup differences (as well as similarities) have yet to be uncovered.

The second limitation revolves around the exclusive use of self-reports across trait and state measures. Self-reports could be subject to—amongst other influences—self-enhancement bias (i.e., social desirability), response styles (e.g., central tendency responding; but see Section 2.6 for details on our data quality control procedures), and restrictions to introspection (e.g., Paulhus & Vazire, 2007; Schimmack et al., 2002). In addition, associations between variables could be inflated by common method variance. Future studies should integrate additional data sources such as informant-reported traits and ESM-based states (e.g., Breil et al., 2019) as well as smartphone sensing (Harari et al., 2020).

The third limitation is that our samples are not representative of the German population. In fact, as detailed in Section 2.4.5, even the convenience samples recruited in Study 2 displayed characteristics (e.g., mostly female and highly educated) that might explain some variance restrictions on more polarizing items and, overall, the tendency toward more socially desirable reports on self- and other-related behavioral evaluations (e.g., more positive views of governmental crisis management) or behavioral changes (e.g., compliance with preventive measures). Possibly, given more diverse samples, we could have observed more variance.

Despite these limitations, we believe that researchers, practitioners, and lay people alike can benefit from the EMOTIONS data and the implications of their analysis. In particular, researchers might engage in (international) collaborative efforts, combining the EMOTIONS data with other (trait and state) data that have been collected world-wide during the COVID-19 pandemic (see Zettler et al., 2022, for an example). This constitutes a fruitful avenue for novel investigations as well as systematic reviews, meta- and mega-analyses.

These pooled insights can inform evidence-based discussions and decision-making processes amongst political and other societal stakeholders. In a similar vein, a lay-person-friendly communication of findings based on EMOTIONS data can educate all members of society on the ways in which (relatively) stable traits and time-varying contextual features may affect how people think, feel, and behave in their daily lives.

We would like to close by focusing on the fact that ESM requires a noteworthy amount of scientific commitment. However, these efforts tend to pay off, producing data with a high degree of detail and analytical potential. As such, "ESM can be a boon to research" (Scollon et al., 2009, p. 27). In this light, we hope that the EMOTIONS data can enrich (psychological) research, socio-political decision-making, and cultural knowledge on psychological dynamics in times of societal crises (and beyond).

## NOTES

1　Note that by using the term *COVID-19 pandemic* (or simply *pandemic*), we refer to the virus (SARS-CoV-2), the infectious disease it causes (COVID-19; WHO, 2022), as well as the regulations and recommendations decreed for its mitigation.

2　We consider decontextualized measures (i.e., measures that do not reference a specific timeframe) as retrospective. For instance, when respondents are asked to indicate their agreement to the statement "I avoid in-person contact with others" (Bierwiaczonek et al., 2020), they ought to consider (and summarize; Paulhus & Vazire, 2007) their past behavior. On these grounds, the methodological limitations of retrospective assessments apply to decontextualized measures too.

3　The terms *state survey* and *ESM survey* are used interchangeably throughout this paper.

4　All other study waves are abbreviated analogously (i.e., Study 1 Wave 2 as S1W2, Study 2 Wave 1 as S2W1, and Study 2 Wave 2 as S2W2).

5　Two data sets were created per study wave because some participants did not complete the ESM phase. Thus, less participants were included in the state data set than in the trait data set (see Table 2). This being said, researchers interested in trait measures only could benefit from greater sample sizes in the trait data sets.

6　Per study wave, the day before the daily number of ESM reports consistently dropped below 100 was considered as cut-off.

7　Mostly, both emails and participant codes were identical. Yet sometimes, two accounts were linked to the same email but slightly different participant codes (that were likely due to a typo; e.g., K05E02H and K05E03H). In this case, the two accounts were merged based on email only. Analogously, if two accounts shared the same participant code but were linked to marginally different emails (that likely belonged to the same person; e.g., honey.bunny@hotmail.com and honey.bunny@gmail.com), they were merged based on participant codes only.

8　We added the variable *wave* to the merged trait data sets to indicate whether a participant completed both waves or only the first/second wave of the respective study. Furthermore, we added the variable *dataset* to the merged state data sets to signify in which study wave a certain row of data was generated.

9　All assumptions of unpaired *t*-tests (i.e., [a] independent samples, [b] bivariate normal distribution, [c] homogeneous variances in both samples; Schober & Vetter, 2019) were met. For details, see the R script on descriptive analyses (to be retrieved from osf.io/6kzx3/).

10 All assumptions of Pearson correlations (i.e., [a] bivariate normal distribution, [b] linear relationship between variables, [c] absence of outliers that were defined as values below 25%-percentile – 3 × interquartile range [IQR] or above 75%-percentile + 3 × IQR, [d] independently measured value pairs; Schober et al., 2018) were met. For details, see the R script on descriptive analyses (to be retrieved from osf.io/6kzx3/).

11 If a trait score existed at a single time point only (e.g., an LM score existing at T1 of S1W1 only), this score substituted the aggregate. Also, please note that since those individuals who did not participate in the ESM phase of S1W2 could not provide an LM score at T4, no *t*-test was computed with data from S1W2.

12 The significance level was set to .050.

13 Diminished degrees of freedom are due to missing values (in this and further instances in the current section).

14 $M_{ESM} = 3.84$, $SD_{ESM} = 0.53$ vs $M_{no\ ESM} = 3.71$, $SD_{no\ ESM} = 0.58$, $t(1297) = -4.01$, $p < .001$, $d = 0.24$ (S2W2)

15 $M_{ESM} = 3.59$, $SD_{ESM} = 0.64$ vs $M_{no\ ESM} = 3.64$, $SD_{no\ ESM} = 0.65$, $t(1297) = 1.15$, $p = .249$, $d = -0.07$ (S2W2)

16 No relationship was found in S2W2, $r(912) = .04$, $p = .214$.

17 Please note that for the sake of brevity, whenever we speak of outliers in the context of the data quality checks, we mean potential outliers only.

18 The latter decision was made because throughout our self-generated items (e.g., coronavirus-related behavioral evaluations, coronavirus exposure), we observed substantially skewed distributions. These would have resulted in low IRVs but should not be mistaken for careless responding.

19 Whenever subtracting 3 × IQR from the 25%-percentile yielded a negative result, which is impossible given that standard deviations are restricted to positive values, we treated IRVs equal to 0 as outliers.

20 The latter two measures subsume a single construct each—self-esteem or Honesty-Humility, respectively—but comprise negatively coded items (Ashton & Lee, 2009; von Collani & Herzberg, 2003).

21 One trait survey was completed per time point (i.e., initial trait survey at T1/T3 and final trait survey at T2/T4). Hence, a participant's completion time was quantified by the time difference between the creation time of the first survey module and the ending time of the last survey module.

22 By reducing the threshold completion time per item to 1 s, we acknowledge that due to the repeated assessment of items across state surveys, training effects that minimize the time required to complete a single state item may have occurred.

23 Illustratively, among the university student samples of Study 1, a male, 67-year-old participant studying psychology in his 6th semester could be easily identified.

24 For illustrative purposes, compare the Pearson correlation between scores on LM and the number of completed state surveys in S1W2, $r(168) = .12$, $p = .107$, with the Pearson correlation between scores on Conscientiousness and the number of completed state surveys in S2W1, $r(1643) = .07$, $p = .004$.

25 E.g., in Study 1, ESM participants' scores on the warm-agreeable (LM) octant of the IAL (Jacobs & Scholl, 2005) were not markedly different from the LM scores of those who left data collection before the ESM phase.

## ADDITIONAL FILES

The additional files for this article can be found as follows:

- **Supplemental Material.** The Supplemental Material includes details on compensation (personalized feedback on emotional well-being), socio-demographic sample information based on the state data set per EMOTIONS study wave (Table S1), frequency distributions of age and household size based on the trait and state data set per study wave (Figures S1 to S4), descriptive statistics on all established trait measures in Study 1 (Table S2) and Study 2 (Table S3) with statistics based on state data sets, as well as details regarding the data quality checks performed per study wave and time point (Table S4).. DOI: https://doi.org/10.5334/jopd.83.s1

- Supplemental data for this article can be found at osf.io/6kzx3/.

## COMPETING INTERESTS

The authors have no competing interests to declare.

## AUTHOR AFFILIATIONS

**Elina Ryvkina**
University of Münster, Germany

**Lara Kroencke** orcid.org/0000-0002-4660-7428
University of Münster, Germany

**Katharina Geukes** orcid.org/0000-0002-7424-306X
University of Münster, Germany

**Julian Scharbert** orcid.org/0000-0003-3020-2976
University of Münster, Germany

**Mitja D. Back** orcid.org/0000-0003-2186-1558
University of Münster, Germany

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

## PEER REVIEW COMMENT

*Journal of Open Psychology Data* has blind peer review, which is unblinded upon article acceptance. The editorial history of this article can be downloaded here:

- **PR File 1.** Peer Review History. DOI: https://doi.org/10.5334/jopd.83.pr1

**TO CITE THIS ARTICLE:**
Ryvkina, E., Kroencke, L., Geukes, K., Scharbert, J., & Back, M. D. (2023). Understanding Psychological Responses to the COVID-19 Pandemic Through ESM Data: The EMOTIONS Project. *Journal of Open Psychology Data,* 11: 6, pp. 1–30. DOI: https://doi.org/10.5334/jopd.83

