## [Peer Review History. · Journal of Open Psychology Data]

Response to Reviewers' Requests

Reviewer 1

Request 1.1: Abstract

The authors might consider defining “ESM” in the abstract before using this abbreviation.

Request 1.2: Drop-out-related interindividual differences

The wording “was unrelated” seems too strong to me against the background of non-significant correlations. Especially non-significant correlations in study 1 could easily be due to differences in statistical power between study 1 and study 2. Overall, I would expect effect sizes to be similar between both studies based on the reported results. Likewise, the significant correlation in study 2 between conscientiousness and number of state reports is very small (and I would expect the effect size for group differences in agreeableness to be also very small). The authors therefore in my opinion currently overstate the significance of this limitation. They might consider including effect sizes for t-tests and mentioning in section 4 that effects were indeed very small.

Request 1.3: Supplementary Material

Markups relating to comments are still included in this document.

Request 1.4: Additional comment regarding possible copyright issues with literal item texts

When inspecting the OSF project, I wondered, however, whether the publication of some item texts on the OSF might be problematic because of copyright issues. For example, for items of the RSES (Rosenberg Self-Esteem Scale) copyrights of the publisher of the corresponding publication might exist. (As these materials are not hosted by JOPD directly I do, however, not think that this should affect the manuscript's suitability for publication in JOPD.)

Reviewer 2

Request 2.1

First, the state surveys and trait measures were mentioned for the first time in the section 2.1 Study Design. When reading this section, I was confused about what exactly the state surveys and trait measures were. The authors did not provide details regarding state surveys and trait measures until very late in section 2.5 Materials. I would suggest the authors provide a high-level table when they mention these key measures for the first time to show what state surveys and trait were measured for each study and each wave (just list survey names and trait questionnaire/subscales names, no need to list all questions in this table).

Otherwise, readers may have to keep this question in mind until the end of the methods section to figure it out.

Request 2.2

Second, if I understand correctly, each wave included three phases for data collection. The first phase (i.e., the first day) was for collecting data for the trait survey. The second phase started from the second day and lasted for 14 days, during which state survey data were collected. In the third and concluding phase, trait survey data was collected again. It is unclear how many days passed between the end of the second phase and the start of the third phase. According to Figure 1, I speculate that the interval is approximately three weeks. Could the authors confirm this and clarify how they decided on this interval? Given that trait is relatively stable, I wonder if five weeks (14 days + ~3 weeks) would be too short to observe any changes in trait measures.

Request 2.3

Third, in Tables 1 and 2, the authors reported the number of state reports per study wave, which is confusing and does not seem an effective way to present the information. For example, by simply reporting the number of state reports, it is unclear which state survey was completed and how many times a survey was completed during the 14 days by each individual. It could be that only a very small part of individuals completed e.g., survey 1, and many individuals completed survey 2, so there would be much missing data in survey 1 but not in survey 2. Similarly, it could be that many individuals completed surveys only on day 1, and a few individuals completed surveys for all 14 days, thus only a small sample of individuals can be used for a longitudinal study. I made these hypothetical examples just to show that all this important information was not able to reflect by reporting the number of state reports. I would suggest the authors make another table to show this important information in an effective way.

Request 2.4

Fourth, in Table 3, the authors reported age, household size, etc. The age range and the household size are wide (16-99 years, 1-99 persons). Within such a wide range, the mean is not a good measure to report. I would suggest the authors report the median instead of the mean. It would be more straightforward if the authors show this information in figures (e.g., age distribution, household size distribution) rather than tables.

Request 2.5

Fifth, I would suggest the authors provide the full names instead of abbreviations when they mention questionnaires or concepts for the first time. This issue exists throughout the paper.